# Accelerating Large Batch Training via Gradient Signal to Noise Ratio (GSNR)

## Abstract

As models for nature language processing (NLP), computer vision (CV) and recommendation systems (RS) require surging computation, a large number of GPUs/TPUs are paralleled with a large batch (LB) to improve training throughput. Training such LB tasks often converges to sharp minimum and downgrades final precision. Adversarial learning (ConAdv) and LANS method scales ImageNet and BERT pretraining up to 96k batch size. In this work, we develop the variance reduced gradient descent technique (VRGD) based on the gradient signal to noise ratio (GSNR) and apply it onto popular optimizers such as SGD/Adam/LARS/LAMB. We carry out a theoretical analysis of VR-SGD's convergence rate to explain its fast training dynamics, and a generalization analysis to demonstrate its smaller generalization gap on LB training. Comprehensive experiments demonstrate that VRGD can remarkably accelerate training ($1.7 \sim 4\times$), narrow the generalization gap and improve final accuracy. We push the batch size limit of BERT pretraining up to 128k/64k and DLRM to 512k without noticeable accuracy loss. We improve ImageNet Top-1 accuracy at 96k by $0.52pp$ than LARS and significantly reduce generalization gap by $68.3\%$.

## 1 Introduction

Recent machine learning models have grown wider and deeper in their architectures (e.g., GPT-3 [Floridi and Chiriatti, 2020], M6 [Lin *et al.*, 2021], Switch Transformer [Fedus *et al.*, 2021]). Training complex models may consume more training data to converge, which needs a surge in computing capacity and efficiency. However, hardware improvement can not keep pace with the expansion of model calculations [Bommasani *et al.*, 2021].

Several techniques to speed up training are proposed. The aggregation and scattering of gradients among massive workers requires an efficient synchronization algorithm. Since the communication bandwidth between GPUs/TPUs is much higher than CPU-GPU (e.g., NVLink, Foley and Danskin [2017]), several efficient synchronization strategies such as Ring-All-Reduce [Gibiansky, 2017] and software toolkits like Horovod [Sergeev and Del Balso, 2018] are proposed to replace the traditional PS-Worker framework [Li *et al.*, 2014b,a]. In addition, training with LB can notably improve throughput [You *et al.*, 2017b; Hoffer *et al.*, 2019]. You *et al.* [2020] successfully train BERT using 1024 TPUs and a LB (64k) within 76 minutes. It demonstrates the efficiency of GPUs/TPUs in large scale parallel tasks. Small batch (SB) is not able to fully utilize those powerful GPUs/TPUs.

However, Keskar *et al.* [2017] theoretically analyze the LB training and finds that it can be easily trapped into sharp local minimum, leading to strong generalization gap. Hoffer *et al.* [2017] indicate that the generalization gap can be attributed to the fewer update steps in LB training compared with SB when using identical epochs. Dai and Zhu [2018] theoretically demonstrate that training with more steps or expanding the learning rate to batch size ratio helps to converge to a flatter local

minimum. Although these issues can be partly resolved by layer-wise adaptive rate scaling (LARS, You *et al.* [2017a]) and layer-wise adaptive large batch (LAMB, You *et al.* [2020]), the batch size limit still exists.

To push the batch size limit and reduce generalization gap, we propose the **element-wise adaptive** techniques called variance reduced gradient descent technique (VRGD) based on GSNR of parameters. Our contributions are listed below:

- We carry out theoretical derivations of convergence rate and generalization analysis to explain why VRGD can accelerate LB training and achieve dramatically smaller generalization gap.

- We perform comprehensive LB experiments and find that VRGD can remarkably accelerate training ($1.7 \sim 4\times$), narrow the generalization gap and improve final precision than previous SOTA (e.g., LAMB, LARS).

- VR-LAMB pushes the batch size limit of BERT pretraining up to **128k/64k** without any accuracy loss, while LAMB stops scaling at 64k/32k. VR-LARS improves the ImageNet Top-1 accuracy to $74.82\%$ at 96k, **0.52pp** higher than LARS. The generalization gap of ImageNet trained with VR-LARS is dramatically reduced by **68.3%** comparing with LARS at 96k. VR-SGD pushes the batch size limit of DLRM from 64k to **512k** without noticeable accuracy loss.

## 2 Related Work

### 2.1 Large Batch Training

Several techniques are proposed to improve the optimization and generalization ability in LB training. Goyal *et al.* [2017] propose a linear scaling rule on learning rate (LR) to achieve the same accuracy as SB and push the batch size limit of ImageNet to 8k. EXTRAP-SGD uses the extra-gradient to stabilize the optimization trajectory and smooth training [Lin *et al.*, 2020]. SWAP quickly trains the model with LB in the first stage and refines it by averaging the weights of multiple SB models in the second stage [Gupta *et al.*, 2020]. Batch Augmentation replicates multiple instances with the same batch size to improve generalization [Hoffer *et al.*, 2019]. The batch size of the experiments in EXTRAP-SGD/SWAP/Batch-Augmentation are less than 8k and are not compared in our experiments.

DecentLaM removes the growing momentum-incurred bias observed in DmSGD and pushes ImageNet to 32k [Yuan *et al.*, 2021]. Layer-wise LRs adjustment optimizers such as LARS [You *et al.*, 2017a], complete layer-wise adaptive rate scaling (CLARS, Huo *et al.* [2021]), LAMB [You *et al.*, 2020] successfully improve the batch size up to 64k both for ImageNet and BERT pretraining without accuracy loss. Recently, the concurrent adversarial learning (ConAdv) method pushes the batch size limit of ImageNet training up to 96k [Liu *et al.*, 2021]. LANS replaces the layer-wise LR adjustment in LAMB with block-wise style [Zheng *et al.*, 2020] and also pushes BERT training up to 96k. Adasum adds those gradients after scaling with proper scalars and even pushes the batch size limit of BERT up to 128k/32k [Maleki *et al.*, 2021].

### 2.2 Gradient Variance and GSNR

Unlike gradient mean, which is widely used in optimizers, gradient variance and its successor GSNR are less used. But gradient variance is frequently discussed in generalization gap. Johnson and Zhang [2013a] propose the stochastic variance reduced gradient (SVRG) with the explicit gradient variance reduction method. Other variants of SVRG like SRVR-NPG, SVRPG and Control Variate methods are also proposed to reduce the gradient variance during training [Liu *et al.*, 2020b; Wang *et al.*, 2013; Papini *et al.*, 2018; Miller *et al.*, 2017]. Rainforth *et al.* [2018] use GSNR to analyze the variational bounds in variational auto-encoder (VAE). McCandlish *et al.* [2018] use GSNR to predict the useful upper bound of batch size. Smith *et al.* [2018]; Devarakonda *et al.* [2017] adaptively increase the batch size during training to achieve acceleration without accuracy loss. Liu *et al.* [2020a] theoretically derive a quantitative relationship between GSNR and generalization gap and prove that larger GSNR leads to better generalization performance. Therefore, gradient variance and GSNR are potentially useful to train deep neural networks.

## 3 Preliminaries

### 3.1 GSNR

Given a data distribution $\mathcal{Z} = \mathcal{X} \times \mathcal{Y}$, a model $\hat{y} = f(x, \theta)$ parameterized by $\theta$ and the loss function $L$. The parameters' gradient *w.r.t.* sample $(x_i, y_i)$ can be written as (Refer to all "notations" in the Appendix.C):

$$\mathbf{g}_i(\theta) := \frac{\partial L(y_i, f(x_i, \theta))}{\partial \theta} \quad (1)$$

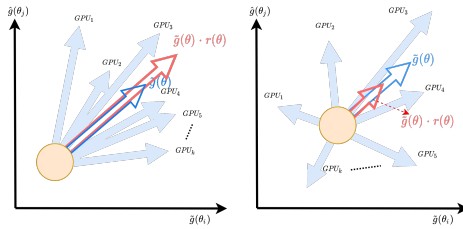

Figure 1: Schematic of VRGD's mechanism: updating parameters with larger GSNR (left panel) and smaller GSNR (right panel).

Then $j$-th parameter' $(\theta_j)$ gradient computed using $(x_i, y_i)$ is $\mathbf{g}_i(\theta_j)$. Here we use $i$ to index the data samples and $j$ to index the parameters of $\theta$. We denote the sample-wise gradient mean as $\tilde{\mathbf{g}}(\theta) = \mathrm{E}_{(x,y)\sim\mathcal{Z}}(\mathbf{g}(x, y, \theta))$ and variance of $\mathbf{g}_i(\theta)$ as $\rho^2(\theta) = \mathrm{Var}_{(x,y)\sim\mathcal{Z}}(\mathbf{g}(x, y, \theta))$. The GSNR for each model parameter $\theta_j$ is defined as:

$$r(\theta_j) := \frac{\tilde{\mathbf{g}}^2(\theta_j)}{\rho^2(\theta_j)} \quad (2)$$

Intuitively, GSNR measures the consistency of the gradient direction of each parameter across a batch of data samples. The gradient space of the parameters tends to converge in the same direction when the GSNR is large, but diverge if the GSNR is small (Figure.1).

### 3.2 GSNR and Generalization Gap

Consider a training set $D = \{(x_1, y_1), ..., (x_n, y_n)\} \sim \mathcal{Z}^{(n)}$, where $n$ samples come from $\mathcal{Z}$, and a test set of dataset size $(n')$ from $\mathcal{Z}'^{(n')}$ denoted by $D' = \{(x'_1, y'_1), ..., (x'_{n'}, y'_{n'})\} \sim \mathcal{Z}'^{(n')}$. The empirical training and test losses can be denoted as:

$$L[D] = \frac{1}{n}\sum_{i=1}^{n} L(y_i, f(x_i, \theta)); \quad L[D'] = \frac{1}{n'}\sum_{i=1}^{n'} L(y'_i, f(x'_i, \theta)) \quad (3)$$

respectively. Then the empirical generalization gap is given by $L[D'] - L[D]$. Both the training loss $L[D]$ and the test loss $L[D']$ would decrease after one training step and can be denoted as $\Delta L[D]$ and $\Delta L[D']$ respectively. The ratio between the expectations of $\Delta L[D]$ and $\Delta L[D']$ for one training step can be denoted as:

$$\mathbf{R}(\mathcal{Z}, n) := \frac{E_{D,D'\sim\mathcal{Z}^n}(\Delta L[D'])}{E_{D\sim\mathcal{Z}^n}(\Delta L[D])} \quad (4)$$

**Assumption 1** (Non-overfitting limit approximation of Liu *et al.* [2020a])**.** *The parameters' gradient over the training set and test set i.e., $\mathbf{g}_D(\theta)$ and $\mathbf{g}_{D'}(\theta)$ obey the same distribution.*

Based on Assumption 1 and using a small learning rate $\lambda \to 0$, Liu *et al.* [2020a] derive the relationship between the one-step generalization ratio (eq.4) and GSNR:

$$\mathbf{R}(\mathcal{Z}, n) = 1 - \frac{1}{n}\sum_j W_j \frac{1}{r_j + \frac{1}{n}}, \quad \text{where } W_j := \frac{E_{D\sim\mathcal{Z}^n}(\Delta L_j[D])}{E_{D\sim\mathcal{Z}^n}(\Delta L[D])} \quad \text{with } \sum_j W_j = 1 \quad (5)$$

where $\Delta L_j[D]$ is the training loss reduction caused by updating $\theta_j$. A more *detailed mathematical derivation* can be found in Liu *et al.* [2020a]. This relationship (eq.5) demonstrates that GSNR $(r_j)$ plays a crucial role in determining the generalization performance of the model. Updating the model parameters with smaller GSNR leads to generalization gap growth. Also note that we have $\mathbf{R}(\mathcal{Z}, n) \to 1$ when $n \to \infty$, which means that training with a larger dataset helps generalization.

## 4 Proposed Algorithms

In this section, we propose VRGD with their general updating rules (taking VR-SGD as an example in Algorithm.1). The SGD is shown in Appendix.D for comparison.

**Algorithm 1:** $VR - SGD$

---

**Input:** require device number $k \geq 2$
**Input:** $B = GlobalBatchSize/k$
**Input:** $\gamma = 0.1$

1 **while** $\theta_t$ *not converged* **do**

    **for** *device* $d = 1$ *to* $k$ **do**

        $\tilde{\mathbf{g}}_d(\theta_t) \leftarrow \frac{1}{B} \sum_{i=1}^{B} \nabla_\theta L(y_i, f(x_i, \theta_{t-1}))$ (Get gradient on each GPU/TPU)

        $\tilde{\mathbf{g}}_d^2(\theta_t) \leftarrow \tilde{\mathbf{g}}_d(\theta_t) \otimes \tilde{\mathbf{g}}_d(\theta_t)$ (Element-wise multiply, so as square terms below)

    $\tilde{\mathbf{g}}(\theta_t) \leftarrow \frac{1}{k} \sum_{d=1}^{k} \tilde{\mathbf{g}}_d(\theta_t)$ (Reduce gradient over all devices)

    $\sigma_t^2 \leftarrow \frac{1}{k} \sum_{d=1}^{k} \tilde{\mathbf{g}}_d^2(\theta_t) - \tilde{\mathbf{g}}^2(\theta_t)$ (Compute gradient variance)

    $r(\theta_t) \leftarrow \frac{\tilde{\mathbf{g}}^2(\theta_t)}{\sigma_t^2}$ (Compute GSNR)

    **for** *layer* $l = 0$ *to* $h$ **do**

        $r(\theta_t^{(l)}) \leftarrow \frac{r(\theta_t^{(l)})}{\frac{1}{J} \sum_{j=1}^{J} r(\theta_{t,j}^{(l)})}$ (Normalize GSNR so that $\overline{r(\theta_t^{(l)})} = 1$)

        $r(\theta_t^{(l)}) \leftarrow \begin{cases} \gamma, & if\ r(\theta_t^{(l)}) < \gamma \\ 1, & if\ r(\theta_t^{(l)}) > 1 \end{cases}$   (Confine the max/min ratio within $\frac{1}{\gamma}$)

    $\theta_t \leftarrow \theta_{t-1} - \lambda \cdot r(\theta_t) \cdot \tilde{\mathbf{g}}(\theta_t)$ (Update weights)

---

## 4.1 VR-SGD's Updating Rules

Consider the simple updating rule for SGD as follows:

$$\theta_t = \theta_{t-1} - \lambda \cdot \tilde{\mathbf{g}}(\theta_t) \tag{6}$$

where $\lambda$ is the learning rate. Previous section demonstrates that updating the weights with larger GSNR confines the model's generalization gap growth during training. Therefore, GSNR can be used in the optimizer for better generalization. In the mathematical derivation of GSNR's role on the generalization gap, all sample-wise gradients for the entire dataset are used to compute the gradient variance, which is less efficient. However, in the LB training training, where each batch is large enough to accurately estimate the gradient variance, we replace the entire dataset with a LB and the sample-wise with device-wise gradient computation. Gradients on each GPU/TPU device can be synchronized using Ring-AllReduce, thus perfectly avoiding the inefficiency of gradient variance computation. The simplified gradient variance computation is as follows:

$$\sigma_t^2 = \frac{1}{k} \sum_{d=1}^{k} \tilde{\mathbf{g}}_d^2(\theta_t) - \tilde{\mathbf{g}}^2(\theta_t) \tag{7}$$

where $k$ devices are used, each of which computes $1/k$ part of the gradient $\tilde{\mathbf{g}}_d(\theta_t)$, the same as what data parallel does. The GSNR can then be easily calculated based on eq.2 ($\rho^2(\theta_j)$ is replaced by $\sigma_j^2$). The mean values of GSNR are removed at each layer before applying gradient to the parameters. This normalization of GSNR ensures that the global averaged GSNR remains at 1.0:

$$r(\theta_t^{(l)}) = \frac{r(\theta_t^{(l)})}{\frac{1}{J} \sum_{j=1}^{J} r(\theta_{t,j}^{(l)})} \tag{8}$$

where $l^{th}$ layer contains $J$ parameters. We constrain the $max/min$ of GSNR within $1/\gamma$ so that those neurons with very small GSNR remain active:

$$r(\theta_t^{(l)}) = \begin{cases} \gamma, & if\ r(\theta_t^{(l)}) < \gamma \\ 1, & if\ r(\theta_t^{(l)}) > 1 \end{cases} \tag{9}$$

where $\gamma$ is a hyper-parameter used here. For simplicity, we **don't tune** $\gamma$ but set it to $0.1$ in all of our experiments by default. Finally, we element-wisely adapt $\lambda$ according to GSNR of each parameter and get the updating rule for VR-SGD:

$$\theta_t = \theta_{t-1} - \lambda \cdot r(\theta_t) \cdot \tilde{\mathbf{g}}(\theta_t) \tag{10}$$

Figure.1 shows the mechanism of VRGD. As for a good estimation of gradient mean (left panel), optimizer should be confident to move along the direction of gradient mean or even further. However, when gradients on the devices are scatteredly distributed (right panel), updating weights with gradient mean may bring noises and slow down convergence, which should be avoided.

**Differences compared with existing LB methods:**

- The linear scaling rule uses the same large LR for all parameters, which tends to diverge when some gradients are too large; LARS/LAMB/LANS use large LRs for some layers but layer-wisely or block-wisely limit LRs when $||\theta_t||$ is compatible with its updating quantity, i.e., $||\theta_t|| \sim ||\lambda \cdot \tilde{\mathbf{g}}(\theta_t)||$; VRGD that we propose here **element-wisely** limit the updating quantity for those parameters without confident gradient estimation (Fig.1b, large gradient variance or small GSNR).

- GSNR and its relationship with generalization gap is discussed in Liu *et al.* [2020a], but further work to embed such GSNR into the optimizers is missing. In our work, we apply GSNR in the SGD/LARS/LAMB and demonstrate that GSNR helps the model maintain a small generalization gap in LB training based on the derivations of the generalization gap and ImageNet experiments.

- VRGD does not need extra-gradient used in EXTRAP-SGD or the two-stage training like SWAP. Sub gradients used in Batch Augmentation have different transforms each while VRGD uses the same transforms. Adasum adaptively sums two gradients scaled by a constant while VRGD still uses the mean gradient.

## 4.2 VR-Adam, VR-LAMB and other VRGD optimizers

GSNR can be easily applied on any optimizer using the general updating rules shown above. Here we discuss those popular optimizers frequently used in the research community, e.g., SGD, Adam, LARS and LAMB. As for VR-Adam, GSNR is calculated directly based on $\tilde{\mathbf{g}}(\theta_t)$ and then used to adapt the gradient mean before gradients' momentum estimation. Similar with the gradients' momentum, we apply the momentum mechanism on GSNR ($\hat{p}_t$) for faster convergence. If we adapt the final update term, i.e. $\theta_t \leftarrow \theta_{t-1} - \lambda \cdot r(\theta_t) \cdot \hat{m}_t/(\sqrt{\hat{v}_t} + \varepsilon)$, the $1^{st}$ and $2^{nd}$ order momentum estimation ($m_t$ and $v_t$) for the next training step would be biased (meaning that the update term cannot be inferred merely on $\hat{m}_t$ and $\hat{v}_t$ since $r(\theta_t) \neq 1$).

VR-LAMB is similar to VR-Adam, except that VR-LAMB layer-wisely adapt the LRs for stable convergence when using very large LRs. VR-Adam and VR-LAMB are shown in Appendix.D. VR-LARS and VR-Momentum, which are based on LARS and Momentum, are similar to VR-SGD that it uses GSNR to adapt the gradient means before applying them to the model weights (algorithms omitted).

# 5 Theoritical Analysis

## 5.1 Convergence Analysis

**Assumption 2** (bounded gradient). $||\nabla L(\theta)|| \leq G$

**Assumption 3** (*l*-smooth). $\exists l > 0$ *satisfies* $||\nabla L(x) - \nabla L(y)|| \leq l||x - y||$

We mathematically derive the convergence rate of VR-SGD under nonconvex settings and assume the training process satisfies Assumption.2 and Assumption.3, which are widely used in convergence analysis [Shamir and Zhang, 2013; Ghadimi and Lan, 2013; Allen-Zhu and Hazan, 2016; Allen-Zhu *et al.*, 2019; You *et al.*, 2020]. Table.1 of Appendix compares the assumptions of ours and those popular optimizers. It shows that our assumptions are weaker than LARS/LAMB/DecentLaM and similar with SGD. Detailed derivations can be found in Appendix.A. Then we have Theorem.1.

**Theorem 1.** *Let* $\lambda_t = \sqrt{\frac{L(\theta_1) - L(\theta^*)}{T||\ell||_1}}$ *and* $\frac{1}{\sqrt{\hat{T}}} = \sqrt{\frac{[(L(\theta_1) - L(\theta^*)]||\ell||_1}{T}}$, *VR-SGD is bounded by:*

$$E||\nabla L(\theta_t)||^2 \leq \mathcal{O}\left((1 + \frac{r_u^2 G^2}{2})\frac{1}{r_l^2\sqrt{\hat{T}}}\right) \tag{11}$$

*where* $r_l$ *and* $r_u$ *are the lower and upper bound of GSNR.*

**Convergence rates discussion**: 1) The convergence rate $\mathcal{O}(\frac{1}{\sqrt{\hat{T}}})$ of VR-SGD is the same as SGD [Johnson and Zhang, 2013b]; 2) VR-SGD's bound depends on the lower ($r_l$) and upper bound ($r_u$) of GSNR. Larger batch size brings smaller gradient variance (eq.43 of Appendix.B) and larger GSNR (both bigger $r_l$ and $r_u$), then may result in **a tighter bound with quicker convergence** (*verified by experiments shown in Figure.2*).

## 5.2 Generalization Gap

This section derives the generalization gap of SGD and VR-SGD during SB and LB scenarios. Detailed derivations can be found in Appendix.B. Citing eq.14 of Liu *et al.* [2020a] below, i.e., when training satisfies Assumption.1 and $\lambda \to 0$, after one training step the expectation of empirical generalization gap at $t^{th}$ step is:

$$E_{D,D'\sim\mathcal{Z}^n}(\Delta_t L[D] - \Delta_t L[D']) = \lambda \sum_j \sigma_{t,j}^2 + O(\lambda^2) \tag{12}$$

where we use $\sigma_{t,j}^2$ and $r_{t,j}$ to denote $\sigma^2(\theta_{t,j})$ and $r(\theta_{t,j})$ for simplicity. Next, we assume that the batch size of LB is $k$ times than that of SB. $\lambda_0$ ($\lambda$) represents the learning rate of SB (LB). The accumulated generalization gap after training $T$ steps for SB using SGD and $T/k$ steps for LB can be derived as follows:

$$E(\mathbf{GAP}_{SB,SGD}) \approx \lambda_0 \sum_{t=1}^{T} \sum_j \sigma_{t,j}^2; \quad E(\mathbf{GAP}_{LB,SGD}) \approx \frac{\lambda}{k} \sum_{t=1}^{T/k} \sum_j \sigma_{t,j}^2 \tag{13}$$

If we assume "$\sigma_{t,j}$ is $t$-independent", eq.13 are simplified as $E(\mathbf{GAP}_{SB,SGD}) \approx \lambda_0 T \sum_j \sigma_j^2$ and $E(\mathbf{GAP}_{LB,SGD}) \approx \frac{\lambda T}{k^2} \sum_j \sigma_j^2$ respectively. Taking $\lambda = k^2\lambda_0$, $E(\mathbf{GAP}_{LB,SGD})$ will have the same accumulated generalization gap as SB. This is known as the linear/square scaling rules. However, the assumption that "$\sigma_{t,j}$ is $t$-independent" is unrealistic. Similarly, the accumulated generalization gap of VR-SGD in LB training can be written as:

$$E(\mathbf{GAP}_{LB,VR-SGD}) \approx \sum_{t=1}^{T/k} \sum_j \frac{\lambda r_{t,j}\sigma_{t,j}^2}{k} = \frac{\lambda}{k} \sum_{t=1}^{T/k} \sum_j \mathbf{g}_{t,j}^2 \tag{14}$$

**The generalization gap of SGD and VR-SGD in LB training:**

When training converges ($\mathbf{g}_{t,j} \to 0$), we have $\mathbf{g}_{t,j}^2 < \sigma_{t,j}^2$ because $r_{t,j} = \mathbf{g}_{t,j}^2/\sigma_{t,j}^2 \to 0$ (verified experimentally by Figure.4 of Liu *et al.* [2020a]). Therefore, we have $\frac{\lambda}{k} \sum_{t=1}^{T/k} \sum_j \mathbf{g}_{t,j}^2 < \frac{\lambda}{k} \sum_{t=1}^{T/k} \sum_j \sigma_{t,j}^2$, i.e., $E(\mathbf{GAP}_{LB,VR-SGD}) < E(\mathbf{GAP}_{LB,SGD})$. This inequality demonstrates that VR-SGD has a **much smaller generalization gap** than SGD in LB training (*verified by our ImageNet experiments shown in Table.3* ).

# 6 Experiments

In this section, we show comprehensive experiments on commonly used LB benchmarks such as BERT Pretraining [Devlin *et al.*, 2019], ImageNet-2012 [Russakovsky *et al.*, 2015] and DLRM [Naumov and Mudigere, 2020]. We mainly adopt the square root rules to scale LRs. We set the hyper-parameters of VRGD as $\gamma = 0.1$ and $k$ to the minimum GPU devices that can hold the LB without out of memory for resource efficiency (but satisfy $k \geq 8$) in all experiments. Similar with other optimizers, VRGD can generate a generally good training curve using default sets. The $1^{st}$ and $2^{nd}$ order decay rates are set to $\beta_1 = \beta_3 = 0.9, \beta_2 = 0.999$ by default. Experiments are performed with TensorFlow on 96 DGX-A100 nodes (768-GPUs).

## 6.1 BERT Pretraining

BERT pretraining is a common NLP task needs speeding up with LB training. For a fair comparison, we use the same settings as LAMB [You *et al.*, 2020] except optimizer and learning rate: (1) BERT large pretrains using Wikipedia and BooksCorpus and then finetunes on SQuAD(v1.1) to evaluate its

precision with F1 score; (2) A two-phase training strategy is used. First $90\%$ steps use a sequence length of 128 (phase-1) and last $10\%$ use a sequence length of 512 (phase-2). Mixed-Batch Training is used when batch size is set to 64k/32k, 96k/32k and 128k/64k.

Table 1: Dev set F1 score of **BERT pretraining and then finetuning on SQuAD(v1.1)**. Each score is the median result of 3 repeated experiments. The baseline of BERT-large on SQuAD(v1.1) is 90.395 [You *et al.*, 2020].

| Batch Size | 16k | 32k | 64k/32k | 64k | 96k/32k | 96k | 128k/32k | 128k/64k |
|---|---|---|---|---|---|---|---|---|
| Steps | 31250 | 15625 | 8599 | 7820 | 6256 | 5214 | 6137 | 4301 |
| LAMB* [You *et al.*, 2020] | 91.35 | 91.48 | 90.58 | - | - | - | - | - |
| Adam* [Nado *et al.*, 2021] | - | 91.58 | 91.04 | 90.46 | - | - | - | - |
| LANS* [Zheng *et al.*, 2020] | - | - | - | - | 90.60 | - | - | - |
| Adasum* [Maleki *et al.*, 2021] | - | - | - | - | - | - | 90.50 | - |
| VR-LAMB | 91.42 | 91.58 | 91.49 | 91.30 | 91.23 | 90.70 | | 90.85 |
| (ours) | (+0.07pp) | (+0.00pp) | (+0.45pp) | (+0.84pp) | (+0.63pp) | | - | |

\* means the F1 scores are cited from their work.
 Using median of repeated experiments is the same as Nado *et al.* [2021].

We use NVIDIA's best practise[1] to carry out VR-LAMB experiments and tune *nothing* of the downstream SQuAD(v1.1) tasks (same as LAMB). Detailed hyper-parameters are listed in Appendix.D. Results shown in Table.1 indicate that:

- VR-LAMB outperforms LAMB (widely used in BERT LB pretraining) in all batch sizes from 16k to 64k/32k. F1 score is improved up to 91.49 at 64k/32k, **0.91pp** higher than LAMB.

- VR-LAMB also outperforms Adam (with standard bias correction and LR discontinuity removal) and LANS by an improvement of **0.84pp** at 64k and **0.63pp** at 96k/32k respectively.

- VR-LAMB pushes the batch size limit up to **128k/64k** using just **4301** steps and maintains a F1 score of 90.85. Although Adasum achieves a F1 score of 90.50 at 128k/32k, but it needs 6137 steps to converge (30% extra steps than VR-LAMB). VR-LAMB achieves 50% less steps than LAMB at 64k/32k and even **0.45pp** higher of F1 score than baseline.

## 6.2 ImageNet with ResNet50

ImageNet training with ResNet50 v1.5 [He *et al.*, 2016a] is a standard CV benchmark for LB training. We use the default sets of official best practise of Google Tensorflow[2] with linear LR warm-up, label smoothing and cosine LR decay (to 0). It is the same setup as LARS [Liu *et al.*, 2021]. We merely adjust the optimizers and learning rate for a fair comparison. We find some successful LB applications using Momentum, LAMB and LARS, but not for Adam, AdaGrad or AdamW optimizers [Goyal *et al.*, 2017; You *et al.*, 2020; Liu *et al.*, 2021]. LARS based on Momentum is more fitful on CV tasks. Therefore, we merely apply VR-LARS on ImageNet. Detailed hyper-parameters are listed in the appendix.D.

Table 2: Top-1 test accuracy of **ImageNet** using ResNet50. Each test accuracy of VR-LARS(ours) is averaged over 5 repeated experiments. The standard Top-1 accuracy of MLPerf-v0.5 is $74.9\%$.

| Batch Size | 2k | 4k | 8k | 16k | 32k | 64k | 96k |
|---|---|---|---|---|---|---|---|
| Momentum* [Goyal *et al.*, 2017] | 76.51% | 76.44% | 76.26% | - | - | - | - |
| DecentLaM* [Yuan *et al.*, 2021] | 76.43% | - | 76.19% | 76.73% | 76.22% | - | - |
| LAMB* [You *et al.*, 2020] | 77.11% | 76.92% | 76.89% | 76.66% | 76.42% | - | - |
| LARS* [Liu *et al.*, 2021] | - | 76.90% | 76.60% | 76.60% | 76.60% | 75.30% | 74.30% |
| VR-LARS | 77.14% | 77.23% | 77.36% | 77.27% | 76.81% | 75.86% | 74.82% |
| (ours) | (+0.03pp) | (+0.31pp) | (+0.47pp) | (+0.54pp) | (+0.21pp) | (+0.56pp) | (+0.52pp) |

\* means the results are cited from their work.

The results shown in Table.2 indicate that:

- VR-LARS outperforms Momentum, DecentLaM, LAMB and LARS (previous SOTA) in all batch sizes (from **0.03pp** to **0.56pp**). The improvements are higher for larger batch size.

---

[1]https://github.com/NVIDIA/DeepLearningExamples/tree/master
[2]https://github.com/tensorflow/models/tree/r1.13.0

251 • VR-LARS achieves $75.86\%$ accuracy at 64k batch size, **0.56pp** higher than LARS. When
252 batch size reaches up to **96k**, VR-LARS maintains $74.82\%$ accuracy, close to the MLPerf-
253 v0.5 standard ($74.9\%$).

254 **Generalization Gap**: Table.3 demonstrates that VR-LARS can dramatically narrow the generalization
255 gap in LB training. The generalization gap is only **1.46** for VR-LARS at 96k (**68.3%** smaller than
256 LARS), even smaller than ConAdv+AA (2.2; Liu *et al.* [2021]). Note that VR-LARS can be used
257 together with ConAdv+AA and other techniques for further improvement.

258

Table 3: Generalization Gap of large batch training on **ImageNet**.

|  | LARS* | | | VR-LARS (ours) | | |
|---|---|---|---|---|---|---|
|  | 32k | 64k | 96k | 32k | 64k | 96k |
| **Train Accuracy** | 82.50 | 79.60 | 78.90 | 80.00 | 78.06 | 76.28 |
| **Test Accuracy** | 76.60 | 75.30 | 74.30 | 76.81 | 75.86 | 74.82 |
| **Generalization Gap** | 5.90 | 4.30 | 4.60 | **3.12** (-47.1%) | **2.20** (-48.8%) | **1.46** (-68.3%) |

\* means the results are cited from [Liu *et al.*, 2021].
Similar phenomenon that train accuracy becomes smaller in VR-LARS is also observed in ConAdv+AA [Liu *et al.*, 2021].

Table 4: Test AUC of **DLRM** trained with SGD and VR-SGD in 1 epoch. The reported results are averaged over 5 repeated experiments. The baseline AUC is $0.8014$ for SGD at 32k batch size.

| Batch Size | 32k | 64k | 128k | 256k | 512k |
|---|---|---|---|---|---|
| SGD† | 0.8014 | 0.8025 | 0.8021 | 0.7827 | 0.7787 |
| VR-SGD (ours) | 0.8026 (+0.12pp) | 0.8048 (+0.23pp) | 0.8042 (+0.21pp) | 0.8023 (+1.96pp) | 0.8013 (+2.26pp) |

† means we reproduce based on NVIDIA's best practise.

## 6.3 DLRM Training

260 Criteo Terabyte click logs dataset (4 billion records) trained with DLRM is a standard CTR prediction
261 benchmark newly added in MLPerf-v0.7. DLRM is used following NVIDIA's best practise[1]. For a
262 fair comparison, we merely modify LRs and optimizers (hyper-parameters are listed in Appendix.D).
263 Settings of Linear LR warm up, polynomial decay and training with 1 epoch are used by their default
264 set up. Results in Table.4 indicates that:

265 • VR-SGD outperforms SGD in all batch size settings. Similar with experiments shown above,
266 the improvement of VR-SGD w.r.t SGD increases along with larger batch sizes (from **0.12pp**
267 to **2.26pp**).

268 • VR-SGD pushes the batch size limit up to 512k and maintains a high AUC of **0.8013**, close
269 to the baseline of 0.8014. Note that Google's submission of MLPerf v0.7 merely uses a
270 maximum batch size of 64k [Kumar *et al.*, 2021].

## 7 Ablation Studies

### 7.1 Orthogonal Experiments

Table 5: Top-1 test accuracy of **CIFAR10** trained with Momentum/Adam/LAMB/LARS optimizers and their corresponding VRGD optimizers using ResNet56. Each test accuracy is averaged over 5 repeated experiments. The reported target accuracy for ResNet56 is $93.03\%$ [He *et al.*, 2016a].

| Batch Size | 256 | 512 | 1k | 2k | 4k | 8k |
|---|---|---|---|---|---|---|
| **Momentum†** | 93.68% | 93.56% | 93.17% | 92.19% | 17.40% | 14.57% |
| **VR-Momentum** (ours) | **93.79%** (+0.11pp) | **93.71%** (+0.15pp) | **93.50%** (+0.33pp) | **93.28%** (+1.09pp) | **92.70%** (+75.30pp) | **90.57%** (+76.00pp) |
| **Adam†** | 91.88% | 92.24% | 92.02% | 91.98% | 59.38% | 20.74% |
| **VR-Adam** (ours) | **92.46%** (+0.58pp) | **92.40%** (+0.16pp) | **92.43%** (+0.41pp) | **92.10%** (+0.12pp) | **91.74%** (+32.36pp) | **90.86%** (+70.12pp) |
| **LAMB†** | 92.08% | 92.03% | 91.90% | 92.13% | 58.35% | 15.13% |
| **VR-LAMB** (ours) | **92.29%** (+0.21pp) | **92.34%** (+0.31pp) | **92.05%** (+0.15pp) | **92.43%** (+0.30pp) | **92.04%** (+33.69pp) | **91.07%** (+75.94pp) |
| **LARS†** | 92.30% | 92.29% | 92.34% | 82.39% | 27.50% | 12.21% |
| **VR-LARS** (ours) | **92.35%** (+0.05pp) | **92.53%** (+0.24pp) | **92.44%** (+0.10pp) | **92.79%** (+10.40pp) | **92.35%** (+64.85pp) | **91.86%** (+79.65pp) |

† means we reproduce based on Google TensorFlow's best practise.

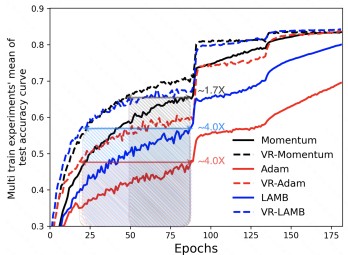

Figure 2: Composite averaged test accuracy or AUC curves of each optimizer for **CIFAR10** experiments. The abrupt surging of accuracy at $91^{th}$ and $136^{th}$ epoch is caused by decaying LR with a rate of $0.1$.

274 In this section, we demonstrate that GSNR is important in optimization and VRGD can be applicable
275 to most popular optimizers using CIFAR10. During CIFAR10 training with ResNet56 [He *et al.*,

2016a,b], we use the default sets of the official best practice for Google Tensorflow[2] and mainly add square-root LR scaling rules to perform the 216 composite experiments shown in Figure.2. Additional linear LR warm-up, label smoothing and cosine LR decay (to 0) techniques are used to stabilize LB training experiments shown in Table.5, the same as ImageNet training. Detailed hyper-parameters are listed in Appendix.D. As for the test accuracy curves, Figure.2 shows the averaged composite test accuracy curve of all 216 experiments for the LR-batch size pairs. Training with VR-Momentum/VR-Adam/VR-LAMB converge much faster ($1.7 \sim 4\times$). As for the final precision, Table.5 demonstrate that VR-Momentum/VR-Adam/VR-LAMB/VR-LARS dramatically outperform Momentum/Adam/LAMB/LARS when batch size is larger than 4k, which demonstrates that VRGD is **applicable** to most popular optimizers in LB training. The improvements of VRGD comparing with their base optimizers grow with the increase of batch size. VRGD optimizers remains convergent when batch size reaches 8k.

## 7.2 GSNR's Behaviour

To understand GSNR's behaviour in VRGD optimizers, we perform the linear regression experiments. The true weights are set to $W_i = i, i \in [1, 10]$ and the corresponding parameters $w_i$ are initialized to zero. Given randomly generated inputs $X$, we have the true labels as $Y = WX$ and the MSE loss as $L = ||Y - wX||_2$. Finally, optimize $w$ with 100 steps.

Training about 50 (half) steps, VR-SGD is able to converge to the test loss where SGD requires 100 steps (Figure.1a of Appendix.D). The weights of VR-SGD (dashed lines of Figure.1b of Appendix.D) converge faster to their ground truth. We find that $w_5, w_6$ converge firstly, then $w_3, w_8$ and finally $w_1, w_{10}$. Consistently, the GSNR of $w_5, w_6$ arise firstly (updating $w_5, w_6$ with larger LRs), then $w_3, w_8$ while the GSNR of $w_5, w_6$ decrease slowly (no need to update the converged weights using large LRs). Finally after step 60, the GSNR of $w_1, w_{10}$ begin to arise. Intuitively, GSNR helps element-wisely fine-tune the LRs for different weights.

## 7.3 Hyper-parameters Sensitivity

There are two main hyper-parameters in VRGD, i.e., normalization strength factor $\gamma$ and the equivalent GPU device number $k$. We take use of linear regression trained with VR-SGD using $batchsize = 2048$ shown above to examine the hyper-parameter sensitivity.

Figure.3 shows that the optimal $\gamma$ is around $(0.04, 0.2)$ for linear regression. Test loss would be larger if $\gamma \to 1$, which means **VR-SGD is reduced to SGD**. It again demonstrates that GSNR is valuable to improve final precision. On the other hand, the optimal $k$ is around $[32, 256]$. This means that each gradient mean calculated using $[8, 64]$ samples on each GPU/TPU device, and gradient variance calculated using $[32, 256]$ values of the gradient mean will return a good evaluation of GSNR. In fact, we do not use the optimal hyper-parameters. Instead, above experiments use $\gamma = 0.1$ and set $k$ to the minimum GPU devices that can hold the LB without out of memory (but satisfy $k \geq 8$, refer all of the hyper-parameters in Appendix.D). Fine-tuning $\gamma$ and $k$ may further improve the results.

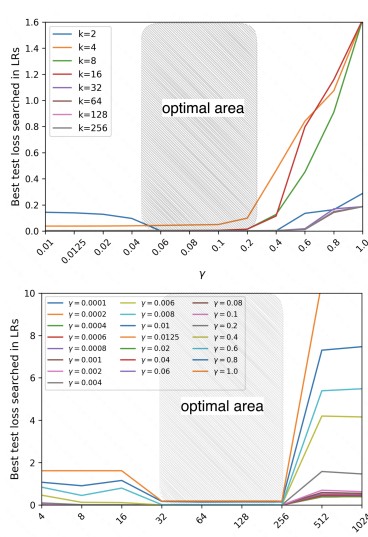

Figure 3: **Hyper parameter sensitivity** experiments: test loss of various $\gamma$ (**Upper panel**) and $k$ (**Bottom panel**).

## 8 Summary

In this paper, we propose the VRGD for large batch training using GSNR. We carry out theoretical derivations of convergence rate and generalization analysis to explain why VRGD can accelerate large batch training and reduce generalization gap. Comprehensive experiments on BERT-pretraining, ImageNet and DLRM verify that VRGD can push the batch size limit than previous SOTA optimizers in LB training and perform better. Codes will be released when published.

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
