# Appendix of Accelerating Large Batch Training via Gradient Signal to Noise Ratio (GSNR)

## A  Proof of VR-SGD's convergence rate:

**Assumption 1** (bounded gradient). $||\nabla L(\theta)|| \leq G$

**Assumption 2** ($l$-smooth). $\exists l > 0$ *satisfies* $||\nabla L(x) - \nabla L(y)|| \leq l||x - y||$

Our assumptions are weaker than LARS/LAMB/DecentLaM and similar with SGD. We assume that the training process satisfies Assumption.1 and Assumption.2, which are widely used in famous optimizers (Table.1). Our derivations does not require the bounded variance. We confine the bound of GSNR to $r(\theta) \leq 1$. GSNR upper bound is not strong and may exist in practise, e.g., Liu *et al.* [2020] found that GSNR will first grow and than decay over time. Layer level bound is also used in the derivations of LARS/LAMB[You *et al.*, 2020] and such assumption can be inferred under the overall bound. For example, if it satisfies $||\nabla L(\theta)|| \leq G_{max}$, then $\exists G_i \leq G_{max}, i \in (1, 2, ..., h)$ satisfy $||\nabla L_i(\theta)|| \leq G_i$. We define the expectation of stochastic mini-batch gradient as the true gradient, i.e., $E(\mathbf{g}^{(i)}) = \nabla_i L(\theta)$.

Table 1: Assumptions comparing with widely used optimizers.

| Optimizer | Asumption.1 (bounded gradient) | Asumption.2 ($l$-smooth) | Other assumption (bounded variance) |
|---|---|---|---|
| SGD[Sa, 2021] | √ | √ | × |
| Adam[Kingma and Ba, 2015] | √ | × | × |
| SVRG[Johnson and Zhang, 2013] | × | √ | × |
| LARS[You *et al.*, 2020] | √ | √ | √ |
| LAMB[You *et al.*, 2020] | √ | √ | √ |
| EXTRAP-SGD[Lin *et al.*, 2020] | × | √ | √ |
| DecentLaM[Yuan *et al.*, 2021] | √ | √ | √ |
| **VR-SGD(ours)** | √ | √ | × |

*Proof.* Inspired by [Sa, 2021; You *et al.*, 2020; Shamir and Zhang, 2013; Ghadimi and Lan, 2013; Allen-Zhu and Hazan, 2016; Allen-Zhu *et al.*, 2019], the convergence of VR-SGD under general nonconvex setting is derived below. VR-SGD's updating rule is:

$$\theta_{t+1}^{(i)} = \theta_t^{(i)} - \lambda_t \cdot r_t^{(i)} \cdot \mathbf{g}_t^{(i)} \tag{1}$$

where $\theta_t^{(i)}$ represents the $i^{th}$ layer parameters of the model at $t^{th}$ training step, $r_t^{(i)}$ and $\mathbf{g}_t^{(i)}$ represents the corresponding GSNR and gradient mean.

From Taylor's theorem and Assumption.2, there exists the upper quadratic bound:

$$L(\theta_{t+1}) \quad \leq L(\theta_t) + \underbrace{< \nabla_i L(\theta_t), \theta_{t+1}^{(i)} - \theta_t^{(i)} >}_{T_1} + \underbrace{\sum_{i=1}^{h} \frac{\ell_i}{2}||\theta_{t+1}^{(i)} - \theta_t^{(i)}||^2}_{T_2} \tag{2}$$

19  The $2^{nd}$ term $T_2$ of eq.(2):

$$T_2 = \sum_{i=1}^{h} \frac{\ell_i}{2} ||\theta_{t+1}^{(i)} - \theta_t^{(i)}||^2 \tag{3}$$

$$= \sum_{i=1}^{h} \frac{\ell_i}{2} (-\lambda_t \cdot r_t^{(i)} \cdot \mathbf{g}_t^{(i)})^2 \tag{4}$$

20  Applying the GSNR definition $r_t^{(i)} := \frac{\mathbf{g}_t^{2(i)}}{\sigma_t^{2(i)}}$, we have:

$$T_2 = \frac{\lambda_t^2}{2} \sum_{i=1}^{h} \ell_i \frac{[\mathbf{g}_t^{(i)}]^6}{[\sigma_t^{(i)}]^4} \tag{5}$$

21  Taking expectation, we have:

$$E(T_2) \leq \frac{\lambda_t^2 r_u^2 G^2 ||\ell||_1}{2} \tag{6}$$

22  The $1^{st}$ term $T_1$ of eq.(2):

$$T_1 = <\nabla_i L(\theta_t), \theta_{t+1}^{(i)} - \theta_t^{(i)}> \tag{7}$$

$$= -\lambda_t \sum_{i=1}^{h} \sum_{j=1}^{d_i} [\nabla_i L(\theta_t)]_j \cdot \frac{[\mathbf{g}_{t,j}^{(i)}]^3}{[\sigma_{t,j}^{(i)}]^2} \tag{8}$$

$$\leq \underbrace{-\lambda_t r_l^2 \sum_{i=1}^{h} \sum_{j=1}^{d_i} \left( [\nabla_i L(\theta_t)]_j \cdot [\mathbf{g}_{t,j}^{(i)}] \right)}_{T_3} \tag{9}$$

$$+ \underbrace{\lambda_t \sum_{i=1}^{h} \sum_{j=1}^{d_i} ([\nabla_i L(\theta_t)]_j \cdot \frac{[\mathbf{g}_{t,j}^{(i)}]^3}{[\sigma_{t,j}^{(i)}]^2}) \mathbf{1}(sign[\nabla_i] \neq sign[\mathbf{g}^{(i)}])}_{T_4} \tag{10}$$

23  where $\mathbf{1}(sign[\nabla_i] \neq sign[\mathbf{g}^{(i)}]) = \mathbf{1}(sign([\nabla_i L(\theta_t)]_j) \neq sign(\mathbf{g}_{t,j}^{(i)}))$.

24  Taking expectation of $T_3$ and $T_4$, we have:

$$E(T_3) = -\lambda_t r_l^2 \sum_{i=1}^{h} \sum_{j=1}^{d_i} E\left( [\nabla_i L(\theta_t)]_j \cdot [\mathbf{g}_{t,j}^{(i)}] \right) \tag{11}$$

$$= -\lambda_t r_l^2 ||\nabla L(\theta_t)||^2 \tag{12}$$

$$E(T_4) = \lambda_t \sum_{i=1}^{h} \sum_{j=1}^{d_i} E[([\nabla_i L(\theta_t)]_j \cdot \frac{[\mathbf{g}_{t,j}^{(i)}]^3}{[\sigma_{t,j}^{(i)}]^2}) \cdot \mathbf{1}(sign([\nabla_i L(\theta_t)]_j) \neq sign(\mathbf{g}_{t,j}^{(i)}))] \tag{13}$$

$$= \lambda_t \sum_{i=1}^{h} \sum_{j=1}^{d_i} E[([\nabla_i L(\theta_t)]_j \cdot \frac{[\mathbf{g}_{t,j}^{(i)}]^3}{[\sigma_{t,j}^{(i)}]^2}) \mid \mathbf{P}(sign([\nabla_i L(\theta_t)]_j) \neq sign(\mathbf{g}_{t,j}^{(i)}))] \tag{14}$$

The probability is bounded by relaxing the condition, then using Markov's and finally Jensen's inequality (inspired by Sign-SGD[Bernstein *et al.*, 2018; Nado *et al.*, 2021]):

$$\mathbf{P}(sign([\nabla_i L(\theta_t)]_j) \neq sign(\mathbf{g}_{t,j}^{(i)})) \tag{15}$$

$$\leq \mathbf{P}\left(|[\nabla_i L(\theta_t)]_j - \mathbf{g}_{t,j}^{(i)}| \geq |[\nabla_i L(\theta_t)]_j|\right) \tag{16}$$

$$\leq \frac{E\left[|[\nabla_i L(\theta_t)]_j - \mathbf{g}_{t,j}^{(i)}|\right]}{|[\nabla_i L(\theta_t)]_j|} \tag{17}$$

$$\leq \frac{\sqrt{E\left[([\nabla_i L(\theta_t)]_j - \mathbf{g}_{t,j}^{(i)})^2\right]}}{|[\nabla_i L(\theta_t)]_j|} \tag{18}$$

$$= \frac{\sigma_{t,j}^{(i)}}{|[\nabla_i L(\theta_t)]_j|} \tag{19}$$

Substituting this relation into $T_4$, we have

$$E(T_4) \leq \lambda_t \sum_{i=1}^{h} \sum_{j=1}^{d_i} E\left[[\nabla_i L(\theta_t)]_j \cdot \frac{[\mathbf{g}_{t,j}^{(i)}]^3}{[\sigma_{t,j}^{(i)}]^2} \cdot \frac{\sigma_{t,j}^{(i)}}{|[\nabla_i L(\theta_t)]_j|}\right] \tag{20}$$

$$\leq \lambda_t \sum_{i=1}^{h} \sum_{j=1}^{d_i} E\left[\frac{[\mathbf{g}_{t,j}^{(i)}]^3}{\sigma_{t,j}^{(i)}}\right] \tag{21}$$

$$\leq \lambda_t r_u^{\frac{1}{2}} G^2 \tag{22}$$

Rearranging eq.(2) and taking expectation, we have:

$$E[L(\theta_{t+1})] \leq E[L(\theta_t)] - \lambda_t r_l^2 ||\nabla L(\theta_t)||^2 + \lambda_t r_u^{\frac{1}{2}} G^2 + \frac{\lambda_t^2 r_u^2 G^2 ||\ell||_1}{2} \tag{23}$$

$$= E[L(\theta_t)] - \lambda_t r_l^2 ||\nabla L(\theta_t)||^2 + \lambda_t r_u^{\frac{1}{2}} G^2 \cdot (1 + \frac{\lambda_t r_u^{\frac{3}{2}} ||\ell||_1}{2}) \tag{24}$$

Summing this until step $T$, we have:

$$E[L(\theta_{T+1})] \leq L(\theta_1) - \lambda_t r_l^2 \sum_{t=1}^{T} ||\nabla L(\theta_t)||^2 + T\lambda_t r_u^{\frac{1}{2}} G^2 \cdot (1 + \frac{\lambda_t r_u^{\frac{3}{2}} ||\ell||_1}{2}) \tag{25}$$

Rearranging this and assuming $\theta^*$ to be the optimal model parameters satisfies $L(\theta^*) \leq E[L(\theta_{T+1})]$:

$$\frac{1}{T} \sum_{t=1}^{T} ||\nabla L(\theta_t)||^2 \leq \frac{1}{r_l^2}\left[\frac{L(\theta_1) - E[L(\theta_{T+1})]}{\lambda_t T} + r_u^{\frac{1}{2}} G^2 \cdot (1 + \frac{\lambda_t r_u^{\frac{3}{2}} ||\ell||_1}{2})\right] \tag{26}$$

$$\leq \frac{1}{r_l^2}\left[\frac{L(\theta_1) - L(\theta^*)}{\lambda_t T} + r_u^{\frac{1}{2}} G^2 \cdot (1 + \frac{\lambda_t r_u^{\frac{3}{2}} ||\ell||_1}{2})\right] \tag{27}$$

Taking $\lambda_t = \sqrt{\frac{L(\theta_1) - L(\theta^*)}{T ||\ell||_1}}$, we can get the bound of VR-SGD:

$$E||\nabla L(\theta_t)||^2 \leq \frac{1}{r_l^2}\left[\sqrt{\frac{[(L(\theta_1) - L(\theta^*)]||\ell||_1}{T}} + r_u^{\frac{1}{2}} G^2 (1 + \frac{r_u^{\frac{3}{2}}}{2}\sqrt{\frac{[(L(\theta_1) - L(\theta^*)]||\ell||_1}{T}})\right] \tag{28}$$

Denoting $\frac{1}{\sqrt{\hat{T}}} = \sqrt{\frac{[(L(\theta_1) - L(\theta^*)]||\ell||_1}{T}}$, we have:

$$E||\nabla L(\theta_t)||^2 \leq \mathcal{O}\left((1 + \frac{r_u^2 G^2}{2})\frac{1}{r_l^2 \sqrt{\hat{T}}}\right) \tag{29}$$

## B  Generalization Gap Derivations of SGD and VR-SGD in LB Scenarios

Inspired by [Liu *et al.*, 2020], the generalization gap of SB and LB using SGD and VR-SGD is derived below. Firstly, the derivations of one step generalization gap are briefly reviewed below and the detailed derivations can be reached in [Liu *et al.*, 2020]. The gradient mean over training set $D$ is denoted as $\mathbf{g}_D(\theta)$. $\mathbf{g}_i(\theta)$ denotes the gradient of a single data sample and $\tilde{\mathbf{g}}(\theta)$ to denote its expectation over the entire data distribution. Similarly we denote $\mathbf{g}_{D'}(\theta)$ as the gradient mean over test set $D'$.

$$\mathbf{g}_D(\theta) = \frac{1}{n} \sum_{i=1}^{n} \mathbf{g}(x_i, y_i, \theta) = \frac{\partial L[D]}{\partial \theta}$$

$$\mathbf{g}_{D'}(\theta) = \frac{1}{n'} \sum_{i=1}^{n'} \mathbf{g}(x_i', y_i', \theta) = \frac{\partial L[D']}{\partial \theta} \tag{30}$$

Using Assumption 0.0.1 in the main context, we have:

$$\mathrm{E}_{D \sim \mathcal{Z}^n}[\mathbf{g}_D(\theta)] = \mathrm{E}_{D, D' \sim \mathcal{Z}^n}[\mathbf{g}_{D'}(\theta)] = \tilde{\mathbf{g}}(\theta) \tag{31}$$

$$\mathrm{Var}_{D \sim \mathcal{Z}^n}[\mathbf{g}_D(\theta)] = \sigma^2(\theta) \tag{32}$$

After one training step, the model parameters are updated by $\Delta \theta = -\lambda \mathbf{g}_D(\theta)$. If $\lambda$ is small enough, the reduction in one-step training and test loss can be approximated as:

$$\Delta L[D] \approx -\Delta \theta \cdot \frac{\partial L[D]}{\partial \theta} + O(\lambda^2)$$

$$= \lambda \mathbf{g}_D(\theta) \cdot \mathbf{g}_D(\theta) + O(\lambda^2) \tag{33}$$

$$\Delta L[D'] \approx -\Delta \theta \cdot \frac{\partial L[D']}{\partial \theta} + O(\lambda^2)$$

$$= \lambda \mathbf{g}_D(\theta) \cdot \mathbf{g}_{D'}(\theta) + O(\lambda^2) \tag{34}$$

Empirically, $\Delta L[D]$ will be larger than $\Delta L[D']$, and the generalization gap will gradually increase. When $\lambda \to 0$, after one single training step the empirical generalization gap is denoted as $\bigtriangledown$ for simplicity. Therefore we have

$$\bigtriangledown := \Delta L[D] - \Delta L[D'] \tag{35}$$

$$\approx \lambda \mathbf{g}_D(\theta) \cdot \mathbf{g}_D(\theta) - \lambda \mathbf{g}_D(\theta) \cdot \mathbf{g}_{D'}(\theta) \tag{36}$$

$$= \lambda (\tilde{\mathbf{g}}(\theta) + \epsilon)(\tilde{\mathbf{g}}(\theta) + \epsilon - \tilde{\mathbf{g}}(\theta) - \epsilon') \tag{37}$$

$$= \lambda (\tilde{\mathbf{g}}(\theta) + \epsilon)(\epsilon - \epsilon') \tag{38}$$

Note that $\epsilon$ and $\epsilon'$ are random variables with zero mean ($E(\epsilon) = E(\epsilon') = 0$) and the variance of $\epsilon$ is $\sigma^2(\theta)$. They are also independent. Therefore the expectation of $\bigtriangledown$ is simplified as:

$$E_{D, D' \sim \mathcal{Z}^n}(\bigtriangledown) = E(\lambda \epsilon \cdot \epsilon) + O(\lambda^2) \tag{39}$$

$$= \lambda \sum_j \sigma^2(\theta_j) + O(\lambda^2) \tag{40}$$

where $\sigma^2(\theta_j)$ is the gradient variance of the parameters $\theta_j$. For simplicity, we use $\sigma_j^2$, $r_j$, and $\mathbf{g}_{D,j}$ to denote $\sigma^2(\theta_j)$, $r(\theta_j)$, and $\mathbf{g}_D(\theta_j)$ respectively.

Next, we denote the empirical generalization gap at $t^{th}$ step as $\bigtriangledown_t$, then we have the accumulated generalization gap after training $T$ steps for SB with SGD:

$$\mathbf{GAP}_{SB, SGD} := \bigtriangledown_1 + \bigtriangledown_2 + ... + \bigtriangledown_T = \sum_{t=1}^{T} \bigtriangledown_t \tag{41}$$

Taking expectation, we have:

$$E(\mathbf{GAP}_{SB, SGD}) = E(\sum_{t=1}^{T} \bigtriangledown_t) = \sum_{t=1}^{T} E(\bigtriangledown_t) \approx \lambda_0 \sum_{t=1}^{T} \sum_j \sigma_{t,j}^2 \tag{42}$$

where $\lambda_0$ denotes the learning rate of SB. As for LB scenarios, we assume the batch size of LB is $k$ times as the SB, then we have the gradient variance of SB and LB are:

$$\sigma^2_{SB}(\theta) = \text{Var}[\frac{1}{B}\sum_{i=1}^{B}\mathbf{g}_i(\theta)] = \frac{1}{B}\rho^2(\theta)$$

$$\sigma^2_{LB}(\theta) = \text{Var}[\frac{1}{kB}\sum_{i=1}^{kB}\mathbf{g}_i(\theta)] = \frac{1}{kB}\rho^2(\theta)$$

(43)

respectively. Similarly, using eq.43, we have the accumulated generalization gap after training $T/k$ steps for LB:

$$E(\mathbf{GAP}_{LB,SGD}) \approx \lambda \sum_{t=1}^{T/k}\sum_{j}\frac{\sigma^2_{t,j}}{k}$$

(44)

If $\sigma_{t,j}$ is $t$ independent, eq.42 and eq.44 are simplified as:

$$E(\mathbf{GAP}_{SB,SGD}) \approx \lambda_0 T \sum_{j}\sigma^2_j$$

(45)

$$E(\mathbf{GAP}_{LB,SGD}) \approx \frac{\lambda T}{k^2}\sum_{j}\sigma^2_j$$

(46)

respectively. Taking $\lambda = k^2 \cdot \lambda_0$, $E(\mathbf{GAP}_{LB,SGD})$ will have the same accumulated generalization gap as SB. This is known as the linear/square scaling rules. However, the assumption that "$\sigma_{t,j}$ is $t$ independent" is unrealistic. Similarly, the accumulated generalization gap of VR-SGD in LB scenarios can be written as:

$$E(\mathbf{GAP}_{LB,VR-SGD}) \approx \sum_{t=1}^{T/k}\sum_{j}\frac{\lambda \cdot r_{t,j} \cdot \sigma^2_{t,j}}{k} = \frac{\lambda}{k}\sum_{t=1}^{T/k}\sum_{j}\mathbf{g}^2_{t,j}$$

(47)

When training converges, $\mathbf{g}_{t,j} \to 0$, $\mathbf{g}^2_{t,j} < \sigma^2_{t,j}$ because $r_{t,j} = \frac{\mathbf{g}^2_{t,j}}{\sigma^2_{t,j}} \to 0$, which has been verified experimentally (see Figure.4 of [Liu *et al.*, 2020]). Therefore, we have:

$$\frac{\lambda}{k}\sum_{t=1}^{T/k}\sum_{j}\mathbf{g}^2_{t,j} < \lambda\sum_{t=1}^{T/k}\sum_{j}\frac{\sigma^2_{t,j}}{k} \ i.e., \ E(\mathbf{GAP}_{LB,VR-SGD}) < E(\mathbf{GAP}_{LB,SGD})$$

(48)

This inequality demonstrates that the generalization ability of VR-SGD is much better than that of SGD.

# C   Notations

| | |
|---|---|
| $\mathcal{Z}$ | A data distribution satisfies $\mathcal{X} \times \mathcal{Y}$ |
| $(x_i, y_i)$ | A single data sample |
| $D$ | Training set consists of $n$ samples drawn from $\mathcal{Z}$ |
| $D'$ | Test set consists of $n'$ samples drawn from $\mathcal{Z}$ |
| $\theta$ | Model parameters, whose components are denoted as $\theta_j$ |
| $\theta^*$ | The optimal model parameters |
| $\mathbf{g}_i(\theta)$ | Parameters' gradient *w.r.t.* a single data sample $(x_i, y_i)$ |
| $\tilde{\mathbf{g}}(\theta)$ or $\tilde{\mathbf{g}}_d(\theta)$ | Mean values of parameters' gradient over a total data distribution *i.e.*, $\mathrm{E}_{s \sim \mathcal{Z}}(\mathbf{g}_i(\theta))$, or gradient over the data on device $d$. |
| $\mathbf{g}_D(\theta)$ | Average gradient over the training dataset, *i.e.*, $\frac{1}{n}\sum_{i=1}^{n}\mathbf{g}_i(\theta)$ |
| $\mathbf{g}_{D'}(\theta)$ | Average gradient over the test dataset, *i.e.*, $\frac{1}{n'}\sum_{i=1}^{n'}\mathbf{g}_i'(\theta)$. Note that, in eq. (30), we assume $n' = n$ |
| $\rho^2(\theta)$ | Variance of parameters' gradient of a single sample, *i.e.*, $\mathrm{Var}_{s \sim \mathcal{Z}}(\mathbf{g}_s(\theta))$ |
| $\sigma^2(\theta)$ | Variance of the average gradient over a training dataset of size $n$, *i.e.*, $\mathrm{Var}_{D \sim \mathcal{Z}^n}[\mathbf{g}_D(\theta)]$ |
| $\sigma_j^2$ | Same as $\sigma^2(\theta_j)$ |
| $r_j$ or $r(\theta_j)$ | Gradient signal to noise ratio (GSNR) of model parameter $\theta_j$ |
| $r(\theta_t^{(l)})$ | GSNR of model parameters $\theta$ on $l^{th}$ layer at $t^{th}$ step |
| $L[D]$ | Empirical training loss, *i.e.*, $\frac{1}{n}\sum_{i=1}^{n}L(y_i, f(x_i, \theta))$ |
| $L[D']$ | Empirical test loss, *i.e.*, $\frac{1}{n'}\sum_{i=1}^{n'}L(y_i', f(x_i', \theta)))$ |
| $\Delta L[D]$ | One-step training loss decrease |
| $\Delta L_j[D]$ | One-step training loss decrease caused by updating one parameter $\theta_j$ |
| $\bigtriangledown$ | One-step generalization gap increment, *i.e.*, $\Delta L[D]$ - $\Delta L[D']$ |
| $\mathbf{R}(\mathcal{Z}, n)$ | One-step generalization ratio (OSGR) for the training and test sets of size $n$ sampled from data distribution $\mathcal{Z}$, *i.e.*, $\frac{E_{D,D' \sim \mathcal{Z}^n}(\Delta L[D'])}{E_{D \sim \mathcal{Z}^n}(\Delta L[D])}$ |
| $\lambda$ | Learning rate |
| $G$ | Upper bound of the gradients *w.r.t* all training samples |
| $\sigma_l^2$ or $\sigma_u^2$ | Lower and upper coordinate bounded variance of the gradients |
| $r_l$ or $r_u$ | Lower and upper bound of the GSNR |
| $\ell_i$ | Upper bound of $\nabla_i^2 L(\theta_t)$ satisfies $u \in \mathcal{R}^d$, $|u^T\nabla_i^2 L(\theta_t)u| \le \ell_i||u||^2$ |
| $T$ | Max training steps |
| $\epsilon$ | Random variables with zero mean and variance $\sigma^2(\theta)$ |
| $\mathbf{GAP}_{SB,SGD}$ | Accumulated generalization gap of SGD during small batch scenario |

 # D  Algorithms and Experiments

---

**Algorithm 1:** $SGD$

---
**Input:** $B = GlobalBatchSize/DeviceNumber(k)$

1 **while** $\theta_t$ *not converged* **do**

    **for** *device* $d = 1$ *to* $k$ **do**

        $\tilde{\mathbf{g}}_d(\theta_t) \leftarrow \frac{1}{B} \sum_{i=1}^{B} \nabla_\theta L(y_i, f(x_i, \theta_{t-1}))$ (Get gradient on each GPU/TPU)

    $\tilde{\mathbf{g}}(\theta_t) \leftarrow \frac{1}{k} \sum_{d=1}^{k} \tilde{\mathbf{g}}_d(\theta_t)$ (Reduce gradient over all devices)

    $\theta_t \leftarrow \theta_{t-1} - \lambda \cdot \tilde{\mathbf{g}}(\theta_t)$ (Update weights)

---

---

**Algorithm 2:** $VR - Adam$

---
**Input:** require device number $k \geq 2$

**Input:** $B = GlobalBatchSize/k$

**Input:** $\gamma_1 = 0.1$

**Input:** $\beta_1, \beta_2 \in [0,1)$ ($1^{st}$ and $2^{nd}$ order decay rates for momentum of gradient)

**Input:** $\beta_3 \in [0,1)$ ($1^{st}$ order decay rates for momentum of GSNR)

1 **while** $\theta_t$ *not converged* **do**

    $m_0 \leftarrow 0$ (Initialize $1^{st}$ order momentum of gradient)

    $v_0 \leftarrow 0$ (Initialize $2^{nd}$ order momentum of gradient)

    $p_0 \leftarrow 0$ (Initialize $1^{st}$ order momentum of GSNR)

    $t \leftarrow 0$ (Initialize train step)

    **for** *device* $d = 1$ *to* $k$ **do**

        $\tilde{\mathbf{g}}_d(\theta_t) \leftarrow \frac{1}{B} \sum_{i=1}^{B} \nabla_\theta L(y_i, f(x_i, \theta_{t-1}))$ (Get gradient on each GPU/TPU)

        $\tilde{\mathbf{g}}_d^2(\theta_t) \leftarrow \tilde{\mathbf{g}}_d(\theta_t) \otimes \tilde{\mathbf{g}}_d(\theta_t)$ (Element-wise multiply, so as square terms below)

    $\tilde{\mathbf{g}}(\theta_t) \leftarrow \frac{1}{k} \sum_{d=1}^{k} \tilde{\mathbf{g}}_d(\theta_t)$ (Reduce gradient over all devices)

    $\sigma_t^2 \leftarrow \frac{1}{k} \sum_{d=1}^{k} \tilde{\mathbf{g}}_d^2(\theta_t) - \tilde{\mathbf{g}}^2(\theta_t)$ (Compute gradient variance)

    $r(\theta_t) \leftarrow \frac{\tilde{\mathbf{g}}^2(\theta_t)}{\sigma_t^2}$ (Compute GSNR)

    **for** *layer* $l = 0$ *to* $m$ **do**

        $r(\theta_t^{(l)}) \leftarrow \frac{r(\theta_t^{(l)})}{\frac{1}{J} \sum_{j=1}^{J} r(\theta_{t,j}^{(l)})}$ (Normalize GSNR so that $\overline{r(\theta_t^{(l)})} = 1$)

        $r(\theta_t^{(l)}) \leftarrow \begin{cases} \gamma_1 & if \ r(\theta_t^{(l)}) < \gamma_1 \\ 1 & if \ r(\theta_t^{(l)}) > 1 \end{cases}$   (Confine the max/min ratio within $\frac{1}{\gamma_1}$)

    $p_t \leftarrow \beta_3 \cdot p_{t-1} + (1 - \beta_3) \cdot r(\theta_t)$ (Update $1^{st}$ order biased momentum of GSNR)

    $\hat{p}_t \leftarrow p_t/(1 - \beta_3^t)$ (Bias correction)

    $\hat{\mathbf{g}}(\theta_t) \leftarrow \hat{p}_t \cdot \tilde{\mathbf{g}}(\theta_t)$ (Adapt gradient mean with GSNR)

    $m_t \leftarrow \beta_1 \cdot m_{t-1} + (1 - \beta_1) \cdot \hat{\mathbf{g}}(\theta_t)$ (Update $1^{st}$ order biased momentum)

    $v_t \leftarrow \beta_2 \cdot v_{t-1} + (1 - \beta_2) \cdot \hat{\mathbf{g}}^2(\theta_t)$ (Update $2^{nd}$ order biased momentum)

    $\hat{m}_t \leftarrow m_t/(1 - \beta_1^t)$ (Bias correction)

    $\hat{v}_t \leftarrow v_t/(1 - \beta_2^t)$ (Bias correction)

    $\theta_t \leftarrow \theta_{t-1} - \lambda \cdot \hat{m}_t/(\sqrt{\hat{v}_t} + \varepsilon)$ (Update weights)

---

---
**Algorithm 3:** *Adam*
---
**Input:** $\beta_1, \beta_2 \in [0,1)$ ($1^{st}$ and $2^{nd}$ order decay rates for momentum)

1 **while** $\theta_t$ *not converged* **do**

    $m_0 \leftarrow 0$ (Initialize $1^{st}$ order momentum of gradient)

    $v_0 \leftarrow 0$ (Initialize $2^{nd}$ order momentum of gradient)

    $t \leftarrow 0$ (Initialize train step)

    **for** *device* $d = 1$ *to* $k$ **do**

        $\tilde{\mathbf{g}}_d(\theta_t) \leftarrow \frac{1}{B}\sum_{i=1}^{B}\nabla_\theta L(y_i, f(x_i, \theta_{t-1}))$ (Get gradient on each GPU/TPU)

    $\tilde{\mathbf{g}}(\theta_t) \leftarrow \frac{1}{k}\sum_{d=1}^{k}\tilde{\mathbf{g}}_d(\theta_t)$ (Reduce gradient over all devices)

    $m_t \leftarrow \beta_1 \cdot m_{t-1} + (1 - \beta_1) \cdot \tilde{\mathbf{g}}(\theta_t)$ (Update $1^{st}$ order biased momentum of gradient)

    $v_t \leftarrow \beta_2 \cdot v_{t-1} + (1 - \beta_2) \cdot \tilde{\mathbf{g}}^2(\theta_t)$ (Update $2^{nd}$ order biased momentum of gradient)

    $\hat{m}_t \leftarrow m_t/(1 - \beta_1^t)$ (Bias correction)

    $\hat{v}_t \leftarrow v_t/(1 - \beta_2^t)$ (Bias correction)

    $\theta_t \leftarrow \theta_{t-1} - \lambda \cdot \hat{m}_t/(\sqrt{\hat{v}_t} + \varepsilon)$ (Update weights)

---
**Algorithm 4:** $VR - LAMB$
---
**Input:** require device number $k \geq 2$

**Input:** $B = GlobalBatchSize/k$

**Input:** $\gamma_1 = 0.1$

**Input:** $\beta_1, \beta_2 \in [0,1)$ ($1^{st}$ and $2^{nd}$ order decay rates for momentum)

**Input:** $\beta_3 \in [0,1)$ ($1^{st}$ order decay rates for momentum of GSNR)

1 **while** $\theta_t$ *not converged* **do**

    $m_0 \leftarrow 0$ (Initialize $1^{st}$ order momentum of gradient)

    $v_0 \leftarrow 0$ (Initialize $2^{nd}$ order momentum of gradient)

    $p_0 \leftarrow 0$ (Initialize $1^{st}$ order momentum of GSNR)

    $t \leftarrow 0$ (Initialize train step)

    **for** *device* $d = 1$ *to* $k$ **do**

        $\tilde{\mathbf{g}}_d(\theta_t) \leftarrow \frac{1}{B}\sum_{i=1}^{B}\nabla_\theta L(y_i, f(x_i, \theta_{t-1}))$ (Get gradient on each GPU/TPU)

        $\tilde{\mathbf{g}}_d^2(\theta_t) \leftarrow \tilde{\mathbf{g}}_d(\theta_t) \otimes \tilde{\mathbf{g}}_d(\theta_t)$ (Element-wise multiply, so as square terms below)

    $\tilde{\mathbf{g}}(\theta_t) \leftarrow \frac{1}{k}\sum_{d=1}^{k}\tilde{\mathbf{g}}_d(\theta_t)$ (Reduce gradient over all devices)

    $\sigma_t^2 \leftarrow \frac{1}{k}\sum_{d=1}^{k}\tilde{\mathbf{g}}_d^2(\theta_t) - \tilde{\mathbf{g}}^2(\theta_t)$ (Compute gradient variance)

    $r(\theta_t) \leftarrow \frac{\tilde{\mathbf{g}}^2(\theta_t)}{\sigma_t^2}$ (Compute GSNR)

    **for** *layer* $l = 0$ *to* $m$ **do**

        $r(\theta_t^{(l)}) \leftarrow \frac{r(\theta_t^{(l)})}{\frac{1}{J}\sum_{j=1}^{J}r(\theta_{t,j}^{(l)})}$ (Normalize GSNR so that $\overline{r(\theta_t^{(l)})} = 1$)

        $r(\theta_t^{(l)}) \leftarrow \begin{cases} \gamma_1 & if\ r(\theta_t^{(l)}) < \gamma_1 \\ 1 & if\ r(\theta_t^{(l)}) > 1 \end{cases}$   (Confine the max/min ratio within $\frac{1}{\gamma_1}$)

    $p_t \leftarrow \beta_3 \cdot p_{t-1} + (1 - \beta_3) \cdot r(\theta_t)$ (Update $1^{st}$ order biased momentum of GSNR)

    $\hat{p}_t \leftarrow p_t/(1 - \beta_3^t)$ (Bias correction)

    $\hat{\mathbf{g}}(\theta_t) \leftarrow \hat{p}_t \cdot \tilde{\mathbf{g}}(\theta_t)$ (Adapt gradient mean with GSNR)

    $m_t \leftarrow \beta_1 \cdot m_{t-1} + (1 - \beta_1) \cdot \hat{\mathbf{g}}(\theta_t)$ (Update $1^{st}$ order biased momentum)

    $v_t \leftarrow \beta_2 \cdot v_{t-1} + (1 - \beta_2) \cdot \hat{\mathbf{g}}^2(\theta_t)$ (Update $2^{nd}$ order biased momentum)

    $\hat{m}_t \leftarrow m_t/(1 - \beta_1^t)$ (Bias correction)

    $\hat{v}_t \leftarrow v_t/(1 - \beta_2^t)$ (Bias correction)

    $\hat{\mathbf{G}}(\theta_t) \leftarrow \hat{m}_t/(\sqrt{\hat{v}_t} + \varepsilon)$

    **for** *layer* $l = 0$ *to* $m$ **do**

        $\theta_t^{(l)} \leftarrow \theta_{t-1}^{(l)} - \lambda \cdot \frac{\phi(||\theta_t^{(l)}||)}{||\hat{\mathbf{G}}(\theta_t)||} \cdot \hat{\mathbf{G}}(\theta_t)$ (Update weights)

---

**Algorithm 5:** $LAMB$

---

**Input:** $\beta_1, \beta_2 \in [0,1)$ ($1^{st}$ and $2^{nd}$ order decay rates for momentum)
**Input:** scaling function $\phi$
1 **while** $\theta_t$ *not converged* **do**
    $m_0 \leftarrow 0$ (Initialize $1^{st}$ order momentum of gradient)
    $v_0 \leftarrow 0$ (Initialize $2^{nd}$ order momentum of gradient)
    $t \leftarrow 0$ (Initialize train step)
    **for** *device* $d = 1$ *to* $k$ **do**
        $\tilde{\mathbf{g}}_d(\theta_t) \leftarrow \frac{1}{B} \sum_{i=1}^{B} \nabla_\theta L(y_i, f(x_i, \theta_{t-1}))$ (Get gradient on each GPU/TPU)
    $\tilde{\mathbf{g}}(\theta_t) \leftarrow \frac{1}{k} \sum_{d=1}^{k} \tilde{\mathbf{g}}_d(\theta_t)$ (Reduce gradient over all devices)
    $m_t \leftarrow \beta_1 \cdot m_{t-1} + (1 - \beta_1) \cdot \tilde{\mathbf{g}}(\theta_t)$ (Update $1^{st}$ order biased momentum)
    $v_t \leftarrow \beta_2 \cdot v_{t-1} + (1 - \beta_2) \cdot \tilde{\mathbf{g}}^2(\theta_t)$ (Update $2^{nd}$ order biased momentum)
    $\hat{m}_t \leftarrow m_t / (1 - \beta_1^t)$ (Bias correction)
    $\hat{v}_t \leftarrow v_t / (1 - \beta_2^t)$ (Bias correction)
    $\hat{\mathbf{g}}(\theta_t) \leftarrow \hat{m}_t / (\sqrt{\hat{v}_t} + \varepsilon)$
    **for** *layer* $l = 0$ *to* $m$ **do**
        $\theta_t^{(l)} \leftarrow \theta_{t-1}^{(l)} - \lambda \cdot \frac{\phi(||\theta_t^{(l)}||)}{||\hat{\mathbf{g}}(\theta_t)||} \cdot \hat{\mathbf{g}}(\theta_t)$ (Update weights)

---

Table 2: Hyper-parameters of **BERT pretraining** with VR-LAMB.

| Batch Size | Steps | Warm-up Steps Phase1 | Phase-1 LR | Phase-1 Acc-steps (k) | Warm-up Steps Phase2 | Phase-2 LR | Phase-2 Acc-steps (k) | F1 Score |
|---|---|---|---|---|---|---|---|---|
| 16k | 31250 | 2800 | 0.0035 | 8 | 280 | 0.0035 | 32 | 91.42 |
| 32k | 15625 | 2800 | 0.0053 | 8 | 280 | 0.0053 | 32 | 91.58 |
| 64k/32k | 8599 | 2000 | 0.007 | 8 | 200 | 0.0045 | 32 | 91.49 |
| 64k | 7820 | 2000 | 0.007 | 8 | 200 | 0.0055 | 64 | 91.30 |
| 96k/32k | 6256 | 1870 | 0.007 | 12 | 200 | 0.00575 | 32 | 91.23 |
| 96k | 5214 | 1870 | 0.007 | 12 | 187 | 0.0055 | 96 | 90.70 |
| 128k/64k | 4301 | 1760 | 0.007 | 16 | 200 | 0.00575 | 64 | 90.85 |

Note that $\gamma = 0.1$ for all batch size. Acc-steps in NVIDIA's code is equivalent to device number $k$.

Table 3: Hyper-parameters of **ImageNet** trained with VR-LARS optimizer on ResNet50.

| Batch Size | Warm-up Epochs | Best LR | Device Number (k) | Test Accuracy |
|---|---|---|---|---|
| 2k | 0.625 | $7 \cdot 2^0$ | 8 | 77.14% |
| 4k | 1.25 | $7 \cdot 2^{0.5}$ | 8 | 77.23% |
| 8k | 2.5 | $7 \cdot 2^1$ | 16 | 77.36% |
| 16k | 5 | $7 \cdot 2^{1.5}$ | 32 | 77.27% |
| 32k | 14 | $7 \cdot 2^2$ | 64 | 76.81% |
| 64k | 40 | 37 | 128 | 75.86% |
| 96k | 41 | 38 | 192 | 74.82% |

Note that $\gamma = 0.1$ for all batch size.

Table 4: Hyper-parameters of **DLRM** trained with SGD and VR-SGD.

| Opt | Batch Size | Warm-up Epochs | LR | Test Accuracy | Opt | Batch Size | Warm-up Epochs | LR | Test Accuracy |
|---|---|---|---|---|---|---|---|---|---|
| SGD | 32k | $1/2^4$ | $2^{3.5}$ | 0.8014 | VR-SGD | 32k | $1/2^4$ | $2^{3.5}$ | 0.8026 |
| | 64k | $1/2^3$ | $2^4$ | 0.8025 | | 64k | $1/2^3$ | $2^4$ | 0.8048 |
| | 128k | $1/2^2$ | $2^{4.5}$ | 0.8021 | | 128k | $1/2^2$ | $2^{4.5}$ | 0.8042 |
| | 256k | $1/2$ | $2^5$ | 0.7827 | | 256k | $1/2$ | $2^5$ | 0.8023 |
| | 512k | $3/4$ | $2^{5.5}$ | 0.7787 | | 512k | $3/4$ | $2^{5.5}$ | 0.8013 |

Note that $\gamma = 0.1$ and device number $k = 8$ for all batch size.

Table 5: Hyper-parameters of **CIFAR10** trained with Momentum/Adam/LAMB/LARS optimizers and their corresponding VRGD optimizers.

| Opt | Batch Size | Warm-up Epochs | LR | Test Accuracy | Opt | Batch Size | Warm-up Epochs | LR | Test Accuracy |
|---|---|---|---|---|---|---|---|---|---|
| Momentum | 256 | $2^2$ | $\frac{128}{2^2\times100}$ | 93.68% | VR-Momentum | 256 | $2^2$ | $\frac{128}{2^2\times100}$ | 93.79% |
| | 512 | $2^{2.5}$ | $\frac{128}{2^{1.5}\times100}$ | 93.56% | | 512 | $2^{2.5}$ | $\frac{128}{2^{1.5}\times100}$ | 93.71% |
| | 1k | $2^3$ | $\frac{128}{2^1\times100}$ | 93.17% | | 1k | $2^3$ | $\frac{128}{2^1\times100}$ | 93.50% |
| | 2k | $2^{3.5}$ | $\frac{128}{2^{0.5}\times100}$ | 92.19% | | 2k | $2^{3.5}$ | $\frac{128}{2^{0.5}\times100}$ | 93.28% |
| | 4k | $2^4$ | $\frac{128}{2^0\times100}$ | 17.40% | | 4k | $2^4$ | $\frac{128}{2^0\times100}$ | 92.70% |
| | 8k | $2^5$ | $\frac{128}{2^{-0.5}\times100}$ | 14.57% | | 8k | $2^5$ | $\frac{128}{2^{-0.5}\times100}$ | 90.57% |
| Adam | 256 | $2^2$ | $\frac{192}{2^3\times1e4}$ | 91.88% | VR-Adam | 256 | $2^2$ | $\frac{192}{2^3\times1e4}$ | 92.46% |
| | 512 | $2^{2.5}$ | $\frac{192}{2^{2.5}\times1e4}$ | 92.24% | | 512 | $2^{2.5}$ | $\frac{192}{2^{2.5}\times1e4}$ | 92.40% |
| | 1k | $2^3$ | $\frac{192}{2^2\times1e4}$ | 92.02% | | 1k | $2^3$ | $\frac{192}{2^2\times1e4}$ | 92.43% |
| | 2k | $2^{3.5}$ | $\frac{192}{2^1\times1e4}$ | 91.98% | | 2k | $2^{3.5}$ | $\frac{192}{2^1\times1e4}$ | 92.10% |
| | 4k | $2^4$ | $\frac{192}{2^0\times1e4}$ | 59.38% | | 4k | $2^4$ | $\frac{192}{2^0\times1e4}$ | 91.74% |
| | 8k | $2^5$ | $\frac{192}{2^{-1}\times1e4}$ | 20.74% | | 8k | $2^5$ | $\frac{192}{2^{-1}\times1e4}$ | 90.86% |
| LAMB | 256 | $2^2$ | $\frac{64}{2^4\times1e3}$ | 92.08% | VR-LAMB | 256 | $2^2$ | $\frac{64}{2^4\times1e3}$ | 92.29% |
| | 512 | $2^{2.5}$ | $\frac{64}{2^{3.5}\times1e3}$ | 92.03% | | 512 | $2^{2.5}$ | $\frac{64}{2^{3.5}\times1e3}$ | 92.34% |
| | 1k | $2^3$ | $\frac{64}{2^3\times1e3}$ | 91.90% | | 1k | $2^3$ | $\frac{64}{2^3\times1e3}$ | 92.05% |
| | 2k | $2^{3.5}$ | $\frac{64}{2^2\times1e3}$ | 92.13% | | 2k | $2^{3.5}$ | $\frac{64}{2^2\times1e3}$ | 92.43% |
| | 4k | $2^4$ | $\frac{64}{2^1\times1e3}$ | 58.35% | | 4k | $2^4$ | $\frac{64}{2^1\times1e3}$ | 92.04% |
| | 8k | $2^5$ | $\frac{64}{2^1\times1e3}$ | 15.13% | | 8k | $2^5$ | $\frac{64}{2^1\times1e3}$ | 91.07% |
| LARS | 256 | $2^2$ | $\frac{896}{2^2\times100}$ | 92.30% | VR-LARS | 256 | $2^2$ | $\frac{896}{2^2\times100}$ | 92.35% |
| | 512 | $2^{2.5}$ | $\frac{896}{2^{1.5}\times100}$ | 92.29% | | 512 | $2^{2.5}$ | $\frac{896}{2^{1.5}\times100}$ | 92.53% |
| | 1k | $2^3$ | $\frac{896}{2^1\times100}$ | 92.34% | | 1k | $2^3$ | $\frac{896}{2^1\times100}$ | 92.44% |
| | 2k | $2^{3.5}$ | $\frac{896}{2^{0.5}\times100}$ | 82.39% | | 2k | $2^{3.5}$ | $\frac{896}{2^{0.5}\times100}$ | 92.79% |
| | 4k | $2^4$ | $\frac{896}{2^0\times100}$ | 27.50% | | 4k | $2^4$ | $\frac{896}{2^0\times100}$ | 92.35% |
| | 8k | $2^5$ | $\frac{896}{2^{-0.5}\times100}$ | 12.21% | | 8k | $2^5$ | $\frac{896}{2^{-0.5}\times100}$ | 91.86% |

Note that $\gamma = 0.1$ and device number $k = 8$ for all batch size.

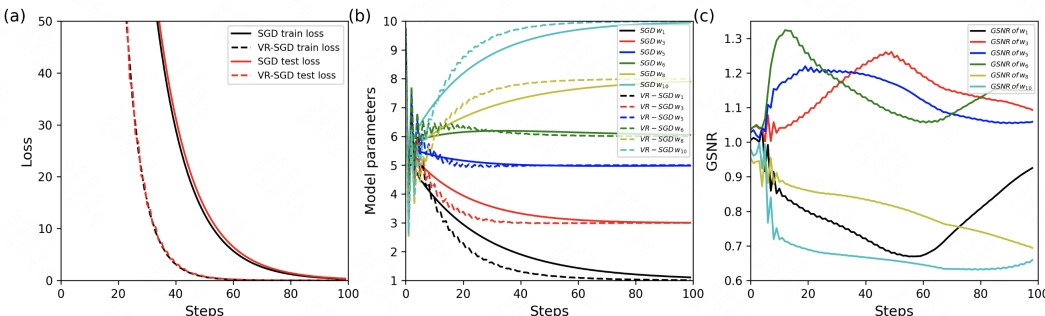

Figure 1: **Linear Regression** experiments trained with SGD and VR-SGD: (a) training and test loss ($batch\ size = 256$); (b) model parameters $w_i, i \in [1, 10]$ ; (c) GSNR of model parameters before $max/min$ constraint used in VR-SGD. Note that $w_i, i \in (2, 4, 7, 9)$ are omitted for simplicity. They behave almost the same as their neighbors.