# OpenReview forum: "Accelerating Large Batch Training via Gradient Signal to Noise Ratio (GSNR)"
_NeurIPS.cc/2023/Conference — Submitted to NeurIPS 2023_

### Official Review · Reviewer_cWbq · 2023-07-01

**Soundness:** 3 good
**Presentation:** 2 fair
**Contribution:** 2 fair
**Rating:** 5
**Confidence:** 2

**Summary:**

this paper proposes a gradient descent technique for learning deep neural nets with large batch sizes. the authors focus on the setting where the model size is small to medium (10M to 300M parameters) and the dataset size is also small to medium (up to 1M images in vision or ~ 3B words in NLP) but with as large batch size as possible. the goal of this works to accelerating the learning of such models(e.g., ResNet on Imagenet, BERT on the original BERT-used corpus) by using less training steps and less training time. The authors were able to show the proposed method can achieve good results (without big drop in eval metrics) while using larger batch sizes than prior arts. And when the proposed method is compared with prior arts at the same batch size, the proposed method seems to outperform prior arts as well.

**Strengths:**

The papers covers standard benchmarks like ResNet on ImageNet and BERT pretraining. and therefore can be fairly compared against many prior arts on the same tasks

**Weaknesses:**

This paper claims "training acceleration" as a key contribution. But throughout the paper, the comparison on speed up is based on number of steps or number of epochs. It is unclear what are the speed advantage in terms of wall-clock time by using the propose technique. I also checked the supp. pdf. In other works (such as LARS and LANs, which are cited by this work), the authors usually report actual wall clock time speedups as they increase the batch size and the compute infra. It was disappointing to not see any mention of that given the authors are using 768 GPUs (therefore I expect very interesting scaling behaviors)

**Questions:**

* how does the proposed method help with the large scale trainings such as CLIP models or DiNOv2 models, which the datasets are much bigger than traditional settings like ImageNet?

* what would be an ideal scenario for using the proposed technique in a computer vision task? from Table 2, there is a clear trade-off between accuracy vs batch size.

* in Table 1 and Table 2, how much time does each training take (under different batch sizes)?

**Limitations:**

the proposed technique is only validated on small-medium size model on small-medium size dataset. it is not applicable (at least no evidence provided) to large scale model training (either large in dataset size or large in model size.

---

> ### Author Rebuttal · Authors · 2023-08-08
>
> We appreciate the reviewer for his/her constructive comments. We carefully address reviewers' questions as follows.
>
> **Q1:** *This paper claims "training acceleration" as a key contribution. But throughout the paper, the comparison on speed up is based on number of steps or number of epochs. It is unclear what are the speed advantage in terms of wall-clock time by using the propose technique. I also checked the supp. pdf. In other works (such as LARS and LANs, which are cited by this work), the authors usually report actual wall clock time speedups as they increase the batch size and the compute infra. It was disappointing to not see any mention of that given the authors are using 768 GPUs (therefore I expect very interesting scaling behaviors)*
>
> Sure, please see Table.1 and Table.2 of the response PDF. Both results show that large batch training largely reduce training time. We add them in the revision.
>
> **Q2:** *how does the proposed method help with the large scale trainings such as CLIP models or DiNOv2 models, which the datasets are much bigger than traditional settings like ImageNet?*
>
> In the CLIP training which is ResNet or Transformer based, they used Adam optimizer and set the batch size to 32k, which can be potentially enlarged with VR-Adam/VR-LARS method we proposed. A larger global batch size means more GPUs can be paralleled to train the model and finally accelerates consuming such bigger training set.
>
> **Q3:** *what would be an ideal scenario for using the proposed technique in a computer vision task? from Table 2, there is a clear trade-off between accuracy vs batch size.*
>
> The ideal scenario is to select the optimal batch size that satisfies desired accuracy and computing time. We suggest to use VR-LARS in computer vision scenario when batch size $\geq$ 2k since VR-LARS is consistently better than any other optimizers listed in Tabel.2 from 2k to 96k.
>
> **Q4:** *in Table 1 and Table 2, how much time does each training take (under different batch sizes)?*
>
> Same as Q1.

---

> ### Comment · Area_Chair_bry1 · 2023-08-18
>
> Dear Reviewer cWbq,
>
> Thanks for your service as a reviewer for the conference.
>
> For this work, you have voted for borderline reject, while most reviewers gave positive ratings to this. Can you please post your post-rebuttal comments? Based on the author's rebuttals and other reviewers' comments, would you like to change your original score? If not it would be great to share your opinions with the authors and the other reviewers so that we—the reviewers and the AC—can reach a consensus.
>
> Best,
> AC

---

> > ### Comment · Reviewer_cWbq · 2023-08-18
> >
> > I ack the rebuttal. I appreciate the PDF response they provided. Im willing to change my vote to borderline accept however I will not change my confidence rating.

---

> > > ### Comment · Area_Chair_bry1 · 2023-08-20
> > >
> > > Dear Reviewer cWbq,
> > >
> > > Thanks for your service as a reviewer for the conference.
> > >
> > > For this work, if you decide to accept it, please change your score. Thanks.
> > >
> > > Best, AC

---

> > > ### Author Response · Authors · 2023-08-21
> > > **Thanks for your comments!**
> > >
> > > As a reminder, can you please change the score to borderline accept  as you said?
> > >
> > > We appreciate your constructive discussion with us.

---

### Official Review · Reviewer_6iTd · 2023-07-05

**Soundness:** 2 fair
**Presentation:** 3 good
**Contribution:** 2 fair
**Rating:** 5
**Confidence:** 3

**Summary:**

The authors strive to propose a heuristic training strategy, called variance reduced gradient descent technique (VRGD), based on the gradient signal to noise ratio, i.e. the ratio between the norm and the variance of gradient. Compared to vanilla training, VRGD scales the learning rate with the gradient signal to noise ratio during each iteration. Then, the authors prove that the proposed method can converge within finite training steps, and claim that it will give better generalization gap compared to the vanilla training. Finally, the authors show the effectiveness of their methods via Bert-training and ImageNet training with up to 96k batch size.

**Strengths:**

***Strengths***

1. The paper is clearly written and easy to follow.

2. With the development of hardwares, training with large batch would gradually become a basic requirement for training large-scale models on gigantic data, such as GPT. In my opinion, the authors are focusing on a very topic worthy to probe.I believe that this paper may have a potential significance not only in academia but also in the industry. However, some modification may be required currently.

3. The authors show some interesting and valuable experiment results, but not comprehensive enough.

**Weaknesses:**

***Weakness***


1. (Major) The authors repeatedly mention that large batch training can lead to sharp minima in Abstract and Introduction, which seems to suggest that the proposed method can avoid such problem. However, I have not found discussions or observations regarding the proposed method can solve this issue. So can the proposed escape these bad minima?

    Considering that many recent works that focuses on guiding training to converge to flat minima i.e. SAM family/gradient norm regularization, I am quite curious what would be like to adopt the proposed method in these algorithms when given large batch training. And from the results, I find that using the proposed method alone would not result better performance than these flat-minima-based methods. I have listed some typical works below.

    [1] Foret, Pierre, et al. "Sharpness-aware minimization for efficiently improving generalization." ICLR2020.

    [2] Kwon, Jungmin, et al. "Asam: Adaptive sharpness-aware minimization for scale-invariant learning of deep neural networks." ICML2021.

    [3] Zhuang, Juntang, et al. "Surrogate gap minimization improves sharpness-aware training." ICLR2021.

    [4] Zhao, Yang, et al. "Penalizing gradient norm for efficiently improving generalization in deep learning." ICML 2022.

2. (Major) Continuing with the previous comment, I find the proposed method is rather heuristic, lacking a clear motivation. I could not find clear rationale and sufficient analysis that the proposed method can benefit training. Essentially, the proposed method is simply to scale the learning rate adaptively based on a specific parameter (Eq. 10). For me, I am not quite convinced that such a learning rate scaling policy will lead to reasonable performance gain. Meanwhile, the author argue that the proposed method can lead to smaller generalization gap. However, the core that supports this claim is based on an empirical observation, which makes the mathematical proof not quite rigorous and the claim much weaker and unhelpful.

3. (Major) I would like to discuss the convergence of the proposed method. Firstly, to my understanding, the convergence analysis focuses on analyzing to what extent can training converge on the given training samples, not testing set. So, it is not quite appropriate to use the convergence curve on the testing set to demonstrate the conclusion regarding the convergence analysis. i.e. Figure 2. Secondly, from Figure 2, the authors state that the proposed method converge 1.7~4 times faster than the conventional optimizers. But, I could not observe such a big gap between them from Figure 2, so could the authors explain how to measure the convergence here. Thirdly, a tighter bound in convergence would not give any information regarding the testing performance. A looser bound and slower convergence rate can give better testing performance in many cases, for example SAM. The author can refer to the paper below.

    [5] Andriushchenko, Maksym, and Nicolas Flammarion. "Towards understanding sharpness-aware minimization." ICML 2022.

    Note that I am not saying a faster convergence is harmful. In my opinion, the core meaning of this convergence section is to prove that the proposed method can converge in finite time. And it is not surprising that The proposed method shares the same convergence rate as that in SGD, i.e. O(1/sqrt{T}) given that this method is to scale the learning rate compared to SGD. In the current version, it appears that the convergence section seems to heavily imply that the proposed method outperforms SGD without a promise of testing performance, which I disagree with.

4. (Minor) Line 32. "However, Keskar et al. [2017] theoretically analyze the LB training and finds that it can be easily trapped into sharp local minimum, leading to strong generalization gap". Actually, the cited paper (Keskar et al. 2017) has not provided theoretical analysis.

5. (Minor) Line 95. To my understanding, the variance of random vectors is a matrix, i.e. covariance matrix. Why is a scalar here?

6. (Minor) It is highly encouraged to show the results of training vision transformers with the proposed methods.

**Questions:**

See weakness.

**Limitations:**

I have not found any discussions about the limitations and potential negative societal impact. But in my opinion, this may not be a problem, since the work only focuses on the learning method in deep learning. Still, it is highly encouraged to add corresponding discussions.

---

> ### Author Rebuttal · Authors · 2023-08-08
>
> We thank the reviewer for his/her detailed and constructive comments. We give the response to address review's concerns step by step below.
>
> **Q1:** *(Major) The authors repeatedly mention that large batch training can lead to sharp minima in Abstract and Introduction, which seems to suggest that the proposed method can avoid such problem...*
>
> Sharp minimum may cause large generalization gap (Foret et al. 2020, Kwon et al. 2021, Zhuang et al. 2022, Zhao et al. 2022, Ahn et al. 2023, Wang et al. 2021, Simsekli et al. 2019). Our current work mainly focuses on generalization gap. Following their work, we add a schematic (Fig.1 in the response PDF) to help understanding how GSNR works to reduce generalization gap. It shows that larger GSNR helps the weights to escape from the large generalization gap area, while smaller GSNR attracts the weights to stay in small generalization gap area. In addition, our experiment shows that the generalization gap was significantly reduced using our method (Table.3, more than 40%).
>
> We are very excited to see other researchers can apply our proposed method on SAM family/gradient norm regularization. Theoretically, they can be combined together to potentially break the current batch size limit. We will try it in the future work.
>
> The main purpose of our work is to push the batch size limit of large batch training without noticeable accuracy loss. We checked the results of SAM family/gradient norm but found that the batch size of their experiments is no more than 4k, which is smaller than our proposed method.
>
> **Reference**
>
> Ahn, K., Jadbabaie, A., and Sra, S. (2023). How to escape sharp minima. arXiv preprint arXiv:2305.15659.
>
> Wang, X., Oh, S., and Rhee, C. H. (2021). Eliminating sharp minima from SGD with truncated heavy-tailed noise. arXiv preprint arXiv:2102.04297.
>
> Simsekli, U., Sagun, L., and Gurbuzbalaban, M. (2019, May). A tail-index analysis of stochastic gradient noise in deep neural networks. In International Conference on Machine Learning (pp. 5827-5837). PMLR.
>
> **Q2:** *(Major) Continuing with the previous comment, I find the proposed method is rather heuristic, lacking a clear motivation. I could not find clear rationale and sufficient analysis that the proposed method can benefit training. Essentially, the proposed method is simply to scale the learning rate adaptively based on a specific parameter (Eq. 10)...*
>
> Here is our logic of motivation:
>
> 1. the literatures pointed out that training with LB may lead to generalization gap
>
> 2. where did the generalization gap come during neural network training?
>
> 3. the literature pointed out updating those weights with small GSNR leads to generalization gap (Liu et al. 2020)
>
> 4. based on LARS/LAMB who are based on learning rate scaling policy, we came up with the idea to update those weights with large GSNR using large learning rate and small GSNR with small learning rate.
>
> Many widely used large batch techniques are based on learning rate scaling policy and actually receive performance gain. For example, LARS/LAMB/LANS used large LRs for the normal layers but layer-wisely or block-wisely limit LRs when $||\theta_t||$ is compatible with its updating quantity and were widely used in research community and industry.
>
> Mathematical proof of generalization gap for our proposed method was shown in Sec 5.2. Additional experiments were further strong supports on our derivations and made the proof results more reliable.
>
> Jinlong Liu, Guoqing Jiang, Yunzhi Bai, Ting Chen, and Huayan Wang. Understanding why neural networks generalize well through GSNR of parameters. In 8th International Conference on Learning Representations, ICLR. OpenReview.net, 2020.
>
>
> **Q3:** *(Major) I would like to discuss the convergence of the proposed method. Firstly, to my understanding, the convergence analysis focuses on analyzing to what extent can training converge on the given training samples, not testing set. So, it is not quite appropriate to use the convergence curve on the testing set to demonstrate the conclusion regarding the convergence analysis. i.e. Figure 2. Secondly, from Figure 2, the authors state that the proposed method...*
>
> Firstly, yes, convergence rate measures how fast optimizer can converge during training while test accuracy measures the generalization.
>
> Secondly, we measured the speed up rates by checking the epochs used to reach the same accuracy. For example, Adam reaches 0.48 accuracy in about 85 epochs while VR-Adam reaches the same accuracy in about 20 epochs, which is a roughly 4 times speed up.
>
> Thirdly, we wanted to express that our proposed method had a tighter bound in large batch scenarios based on theoritical derivations. Testing performance was further compared in the experiment section.
>
> **Q4:** *(Minor) Line 32. "However, Keskar et al. [2017] theoretically analyze the LB training and finds that it can be easily trapped into sharp local minimum, leading to large generalization gap". Actually, the cited paper (Keskar et al. 2017) has not provided theoretical analysis.*
>
> Yes, "theoretically" is removed.
>
> **Q5:** *(Minor) Line 95. To my understanding, the variance of random vectors is a matrix, i.e. covariance matrix. Why is a scalar here?*
>
> $\rho^2(\theta_j)$ is not a scalar. We used $j$ to index the weights.
>
> **Q6:** *(Minor) It is highly encouraged to show the results of training vision transformers with the proposed methods.*
>
> Thanks for the suggestion but we had performed BERT, ImageNet and DLRM for three commonly used scenarios, which are more than popular LARS (Imagenet only) and LAMB (Imagenet and BERT only). We discuss applying on ViT in the discussion section and leave the experiment to future work.

---

> > ### Comment · Reviewer_6iTd · 2023-08-12
> > **Thanks for the kindly response.**
> >
> > Thanks for the kindly response and all the authors' effort. I think the proposed method is interesting, and believe it may have potential impact in learning method. However, I think the current version is not ready to publish. Based on the authors' response, I decide to keep my rate.
> >
> > 1. I can understand the authors' intention. The uploaded pdf again has conveyed the idea that the proposed method would encourage training towards flat minima.However, firstly, I could not find sufficient view regarding why the proposed method would avoid sharp minima, especially considering that unlike SAM, the proposed method does not target to guide the training towards flat minima in a straightforward manner.  Importantly, given the method targets to improve generalization from the perspective of flat minima training, the authors have not provided any comparison or show the compatibility with the SAM family learning methods, which is considered as the current dominating method in regards to this problem. Meanwhile, based on the experiments and my own training experience, the proposed method would not give better results compared to SAM family (96.52 with SAM using ResNet18 on Cifar10 [1], the best acc is 93.79 with the authors' method using ResNet56). And the authors argue in their rebuttal that "We checked the results of SAM family/gradient norm but found that the batch size of their experiments is no more than 4k, which is smaller than our proposed method." However, this does not mean that SAM could not apply to large batch training. Several papers have already presents the results of SAM training with large batch. For example, [2] provides results of SAM training with up to 32k batch size, against what the authors are arguing. So, in my opinion, a basic comparison between the proposed method and SAM is minimum for acceptance.
> >
> >     [1] Efficient Sharpness-aware Minimization for Improved Training of Neural Networks, ICLR2022
> >
> >     [2] Towards Efficient and Scalable Sharpness-Aware Minimization, CVPR2022.
> >
> > 2. Essentially, the proposed method is simply to scale the learning rate adaptively based on a specific parameter (Eq. 10). The core that supports why such training can give better generalization gap is based on an empirical observation, which is quite not rigorous and useful. As far as I know, the mathematical discussion of generalization bound should only be based on some reasonable basic assumptions, like L-smoothness, etc. Meanwhile, the authors have not give clear response in their rebuttal.
> >
> > 3. Based on the results, I could observe that the proposed method presents faster convergence rate for the given case, but the metric that the authors use to gauge the convergence rate is questionable. The authors use the epochs that different optimizers reach the same acc in the middle of the training (in the authors' rebuttal, optimizers reach 0.48 acc), as the speed of convergence. To my understanding, convergence rate is the speed that reaches optimal solution. However, the authors' metric firstly uses the middle state of training, not the optimal solution, and secondly, has not gauged the "speed", i.e. variations of values.

---

> > > ### Author Response · Authors · 2023-08-13
> > > **We appreciate the reviewer's patience and constructive discussion.**
> > >
> > > **Q1:** *I can understand the authors' intention. The uploaded pdf again has conveyed the idea that the proposed method would encourage training towards flat minima.However, firstly, I could not find sufficient view regarding why the proposed method would avoid sharp minima, especially considering that unlike SAM, the proposed method does not target to guide the training towards flat minima in a straightforward manner. Importantly, given the method targets to improve generalization from the perspective of flat minima training, the authors have not provided any comparison or show the...*
> > >
> > > Firstly, we didn't show more supporting results because our proposed method is mainly based on generalization gap, not a straightforward method to escape sharp minimum like SAM. Our proposed method focused on scaling batch size by controlling the generalization gap using GSNR. Therefore we only stated that it helped reducing generalization gap, not from a straightforward way to escape from the sharp minimum. [6] has theoretically showed how GSNR influences generalization gap. We further derived that our proposed method based on GSNR can control generalization gap in large batch scenarios in Sec.5.2 and verified our derivations with ImageNet experiments in Table.3.
> > >
> > > Secondly, we apologize for the misleading but we didn't mean SAM [1] can not be used in large batch training. Instead, the more complex hybrid optimizer is our future work.
> > >
> > > Thirdly, sure, we compare our proposed method with SAM family/Gradient norm on ImageNet-1k with ResNet50 below (these results are cited from [2,3,4,5]). Results show that our proposed method performs better than SAM/ASAM/ESAM/Gradient norm and the same as GSAM. We will add this table in the revision.
> > >
> > > | Batch Size | 512 | 4k |
> > > | --- | --- | --- |
> > > | baseline w.o. SAM | 75.8%[2] | 76.0%[3] |
> > > | SAM (90 epochs) | - | 76.9%[3] |
> > > | SAM (100 epochs) | 76.4%[2] | - |
> > > | ASAM (100 epochs) | 76.6%[2] | - |
> > > | GSAM (90 epochs) | - | 77.2%[3] |
> > > | ESAM (90 epochs) | 77.1%[5] | - |
> > > | Gradient Norm (100 epochs) | - | 77.1%[4] |
> > > | **Ours** (90 epochs) | - | 77.2% |
> > >
> > > **Reference**
> > >
> > > [1] Foret, Pierre, et al. "Sharpness-aware minimization for efficiently improving generalization." ICLR2020.
> > >
> > > [2] Kwon, Jungmin, et al. "Asam: Adaptive sharpness-aware minimization for scale-invariant learning of deep neural networks." ICML2021.
> > >
> > > [3] Zhuang, Juntang, et al. "Surrogate gap minimization improves sharpness-aware training." ICLR2021.
> > >
> > > [4] Zhao, Yang, et al. "Penalizing gradient norm for efficiently improving generalization in deep learning." ICML 2022.
> > >
> > > [5] Efficient Sharpness-aware Minimization for Improved Training of Neural Networks, ICLR2022
> > >
> > > [6] Understanding why neural networks generalize well through GSNR of parameters, ICLR2020.
> > >
> > > **Q2:** *Essentially, the proposed method is simply to scale the learning rate adaptively based on a specific parameter (Eq. 10). The core that supports why such training can give better generalization gap is based on an empirical observation, which is quite not rigorous and useful. As far as I know, the mathematical discussion of generalization bound should only be based on some reasonable basic assumptions, like L-smoothness, etc. Meanwhile, the authors have not give clear response in their rebuttal.*
> > >
> > > The assumptions used to derive generalization gap (Sec 5.2) was Assumption.1 (Non-overfitting limit approximation) and $\lambda\rightarrow$ 0. Note that Assumption.1 was cited from [1] and could hold using early stop strategy. Learning rate assumption can also hold with a commonly used learning rate decay policy. In fact, learning rate was decaying to 0 in our ImageNet experiments.
> > >
> > > We apologize that our previous response was not clear. Our logic is that we first mathematically derived the generalization gap of our proposed method, then we further verify the derivations with experiments. We think such logic is reasonable and helpful.
> > >
> > > **Reference**
> > >
> > > [1] Understanding why neural networks generalize well through GSNR of parameters, ICLR2020.
> > >
> > > **Q3:** *Based on the results, I could observe that the proposed method presents faster convergence rate for the given case, but the metric that the authors use to gauge the convergence rate is questionable. The authors use the epochs that different optimizers reach the same acc in the middle of the training (in the authors' rebuttal, optimizers reach 0.48 acc), as the speed of convergence. To my understanding, convergence rate is the speed that reaches optimal solution. However, the authors' metric firstly uses the middle state of training, not the optimal solution, and secondly, has not gauged the "speed", i.e. variations of values.*
> > >
> > > Yes, the reviewer is right. We recompute with the optimal position and the revised version is $1\sim2\times$.

---

> > > > ### Comment · Area_Chair_bry1 · 2023-08-20
> > > >
> > > > Dear Reviewer 6iTd,
> > > >
> > > > Thanks for your service as a reviewer for the conference.
> > > >
> > > > For this work, it has a borderline score. Can you please post your post-rebuttal comments? Based on the author's rebuttals and other reviewers' comments, would you like to change your original score? If not, please share your opinions with the authors and the other reviewers, which helps AC and reviewers can reach a consensus.
> > > >
> > > > Best, AC

---

> > > > > ### Comment · Reviewer_6iTd · 2023-08-20
> > > > >
> > > > > Thanks for the authors' kindly response. After the discussions, the authors address my concerns to certain extent. However, this paper still need to be more clear in regards to the motivation and maths. I decide to raise my score to borderline accept. Thanks for the authors' effort in addressing my concerns.

---

### Official Review · Reviewer_FtDV · 2023-07-06

**Soundness:** 3 good
**Presentation:** 3 good
**Contribution:** 3 good
**Rating:** 5
**Confidence:** 4

**Summary:**

This paper examines the improvement of training throughput in a large batch parallel training setting. By employing the gradient signal to noise ratio (GSNR) as a measurement of the generalization gap during training, the authors introduce a variance-reduced gradient descent method designed for large batch training scenarios. The authors provide theoretical analysis to substantiate that the proposed variance-reduced gradient descent (VRSGD) method exhibits superior generalization compared to stochastic gradient descent (SGD) and potentially achieves faster convergence. Additionally, experimental evaluations on BERT pretraining, ResNet training, and DLRM training are presented to demonstrate the superiority of the proposed method over alternative approaches for large batch training. Furthermore, orthogonal experiments and analyses pertaining to the behavior of GSNR and sensitivity to hyperparameters are conducted to further support the superiority of the proposed method.

**Strengths:**

1. The research topic of large batch training is of considerable interest.

2. The proposed method is both simple and effective for large-batch training.

3. The theoretical analyses of the convergence rate and generalization are persuasive.

4. The proposed method consistently demonstrates improvements over other baseline approaches.

5. The paper is well-written and easy to understand.

**Weaknesses:**

1. The contribution is limited. It seems that the authors just introduce GSNR into the large batch training, but do not highlight why GSNR is important to large batch training.

2. Although the theoretical analysis and experimental results presented in the study are persuasive, the rationale behind the necessity of the GSNR method for large-batch training remains unclear. It is essential to provide a more comprehensive explanation of why GSNR is specifically relevant and advantageous in the context of large batch training. While GSNR can potentially enhance generalization in various settings, a more explicit justification is required to elucidate its particular suitability and effectiveness for large-batch training scenarios.

**Questions:**

NA

---

> ### Author Rebuttal · Authors · 2023-08-08
>
> We thank the reviewer for his/her detailed and constructive comments. We give the response to address review's concerns step by step below.
>
> **Q:** *The contribution is limited. It seems that the authors just introduce GSNR into the large batch training, but do not highlight why GSNR is important to large batch training. Although the theoretical analysis and experimental results presented in the study are persuasive, the rationale behind the necessity of the GSNR method for large-batch training remains unclear. It is essential to provide a more comprehensive explanation of why GSNR is specifically relevant and advantageous in the context of large batch training. While GSNR can potentially enhance generalization in various settings, a more explicit justification is required to elucidate its particular suitability and effectiveness for large-batch training scenarios.*
>
> A more explicit justification to elucidate why our proposed method can work in large batch scenarios is shown below:
>
> 1. **Methodological perspective.** Our proposed method provides an element-wise level of learning rate adjustment that is more accurate than existing methods and becomes more accurate when batch size gets larger. The linear scaling rule uses the same large LR for all parameters. LARS/LAMB/LANS use large LRs for the normal layers but layer-wisely or block-wisely limit LRs when $||\theta_t||$ is compatible with its updating quantity. VRGD that we proposed **element-wisely** limits the updating quantity for those parameters without confident gradient estimation (Fig.1b in the main context, large gradient variance or small GSNR). GSNR estimation becomes more accurate when batch size is larger. Therefore, when batch size gets extremely large, such mechanism to stabilize training may become even more accurate and helpful.
>
> 2. **Convergence rate perspective.** Applying our proposed method on basic optimizers may make the upper bound of convergence rate much tighter when increasing the batch size. For example, VR-SGD's bound depends on the lower ($r_l$) and upper bound ($r_u$) of GSNR. Larger batch size brings smaller gradient variance (eq.43 of Appendix.B) and larger GSNR (both bigger $r_l$ and $r_u$), then may result in **a tighter bound with quicker convergence** (*verified by experiments*).
>
> 3. **Generalization gap perspective.** Our proposed method can reduce more generalization gap when batch size is larger. Based on the derivations in Sec 5.2, VR-SGD has a **much smaller generalization gap** than SGD in LB training (*verified by our ImageNet experiments shown in Table.3 of the main context*). When scaling up the batch size, such mechanism to reduce generalization becomes even more useful. Table.3 of the the main context shows that generalization gap drops 47.1% at 32k, 48.8% at 64k and 68.3% at 96k.
>
> 4. **GSNR effectiveness perspective.** The theoretical explanation of the mechanism how updating weights with smaller GSNR brings generalization gap is comprehensively discussed in previous study (Liu et al.2020). We further carried out many ablation studies in Sec.7 and found that final accuracy drops in large batch training without GSNR, which demonstrates its effectiveness in large batch scenarios.
>
> Furthermore, we add Fig.1 in the response PDF. Following previous discussion on generalization gap (Foret et al. 2020, Kwon et al. 2021, Zhuang et al. 2022, Zhao et al. 2022, Zhao et al. 2023, Wang et al. 2021, Simsekli et al. 2019), we add a schematic to help understanding how GSNR works to reduce generalization gap. It shows that larger GSNR helps the weights to escape from the large generalization gap area, while smaller GSNR attracts the weights to stay in small generalization gap area.
>
>
> **Reference**
>
> Jinlong Liu, Guoqing Jiang, Yunzhi Bai, Ting Chen, and Huayan Wang. Understanding why neural networks generalize well through GSNR of parameters. In 8th International Conference on Learning Representations, ICLR. OpenReview.net, 2020.
>
> Foret, P., Kleiner, A., Mobahi, H., and Neyshabur, B. (2020). Sharpness-aware minimization for efficiently improving generalization. arXiv preprint arXiv:2010.01412.
>
> Kwon, J., Kim, J., Park, H., and Choi, I. K. (2021, July). Asam: Adaptive sharpness-aware minimization for scale-invariant learning of deep neural networks. In International Conference on Machine Learning (pp. 5905-5914). PMLR.
>
> Zhuang, J., Gong, B., Yuan, L., Cui, Y., Adam, H., Dvornek, N., ... and Liu, T. (2022). Surrogate gap minimization improves sharpness-aware training. arXiv preprint arXiv:2203.08065
>
> Zhao, Y., Zhang, H., and Hu, X. (2022, June). Penalizing gradient norm for efficiently improving generalization in deep learning. In International Conference on Machine Learning (pp. 26982-26992). PMLR.
>
> Ahn, K., Jadbabaie, A., and Sra, S. (2023). How to escape sharp minima. arXiv preprint arXiv:2305.15659.
>
> Wang, X., Oh, S., and Rhee, C. H. (2021). Eliminating sharp minima from SGD with truncated heavy-tailed noise. arXiv preprint arXiv:2102.04297.
>
> Simsekli, U., Sagun, L., and Gurbuzbalaban, M. (2019, May). A tail-index analysis of stochastic gradient noise in deep neural networks. In International Conference on Machine Learning (pp. 5827-5837). PMLR.

---

> > ### Comment · Reviewer_FtDV · 2023-08-20
> > **Official Comment by Reviewer FtDV**
> >
> > Thank you for the authors' response. Following a thorough review of the rebuttal, some of my initial concerns regarding the significance of the GSNR in the context of large-batch training have been alleviated to a certain extent. It is now evident that GSNR is helpful for generalization and convergence when dealing with larger batch sizes. In light of this clarifying information, I have chosen to retain my original evaluation score.

---

### Official Review · Reviewer_Xfpk · 2023-07-07

**Soundness:** 3 good
**Presentation:** 2 fair
**Contribution:** 3 good
**Rating:** 7
**Confidence:** 2

**Summary:**

The paper proposes a new method for large batch training. It is based on the insight that the gradient-to-signal-noise-ratio for each parameter should be reflected in its learning rate, and hence modifies gradient descent to reduce the variance of the gradients. The paper then shows convergence rates of VRSGD, which are the same as SGD asymptotically, and states that VR-SGD is particularly suited to large batch training where the GSNR will be high. The paper also shows that the proposed method has a smaller generalization gap than SGD in the large batch setting. Experiments are then performed to support these claims on a variety of tasks and architectures.

**Strengths:**

1. The method is based on a solid insight.
2. The empirical results indicate decent gains over baselines.

**Weaknesses:**

1. The assumptions of smoothness and bounded gradients seem a bit too strong.

**Questions:**

Can the assumptions for the theoretical analysis be relaxed?

**Limitations:**

See weaknesses.

---

> ### Author Rebuttal · Authors · 2023-08-08
>
> We thank the reviewer for his/her detailed and constructive comments. We give the response to address review's concerns below.
>
> **Q:** *The assumptions of smoothness and bounded gradients seem a bit too strong. Can the assumptions for the theoretical analysis be relaxed?*
>
> Sure, the bounded gradients assumption can be relaxed. Moulines et al. 2011 (their Theorem.4) and Nguyen et al. 2018 (their Lemma.2) derived that SGD can still be bounded without bounded gradients assumption, but they still needed the $l$-smooth assumption.
>
> Based on the derivations of Theorem.1 in Johnson and Zhang (2013), we can derive a similar bound for our proposed method without bounded gradients assumption by taking $\lambda=\frac{1}{\sqrt{T}}$, we have
>
> $$\mathbb{E}||\nabla L(\theta_t)||^2 \leq \left[ \frac{1}{\gamma r_l (\sqrt{T}-2l)} + \frac{2l}{\sqrt{T}-2l} \right]^2 \mathbb{E}[L(\theta_t) - L(\theta_*)]$$
>
> Therefore, when $\lambda$ gradually decreases with $T$, our proposed method still converges with $O(\frac{1}{\sqrt{T}})$ without bounded gradient assumption. We add this part in the appendix.
>
> Note that most previous optimizers used the same or stronger assumptions than ours. Table.1 in the Appendix shows that our assumptions are weaker than LARS/LAMB/DecentLaM and the same as common SGD.
>
> **Reference**
>
> Moulines, E., and Bach, F. (2011). Non-asymptotic analysis of stochastic approximation algorithms for machine learning. Advances in neural information processing systems, 24.
>
> Johnson, R., and Zhang, T. (2013). Accelerating stochastic gradient descent using predictive variance reduction. Advances in neural information processing systems, 26.
>
> Nguyen, L., Nguyen, P. H., Dijk, M., Richtárik, P., Scheinberg, K., and Takác, M. (2018, July). SGD and Hogwild! convergence without the bounded gradients assumption. In International Conference on Machine Learning (pp. 3750-3758). PMLR.

---

> > ### Comment · Reviewer_Xfpk · 2023-08-11
> > **Response**
> >
> > I thank the authors for their response, and stand by my review.

---

### Official Review · Reviewer_xC4Q · 2023-07-11

**Soundness:** 3 good
**Presentation:** 2 fair
**Contribution:** 2 fair
**Rating:** 5
**Confidence:** 5

**Summary:**

This paper focus on using large-batch training to accelerate the training of neural network. Specifically, the authors try to use variance reduced gradient descent technique to scale up the batch size and therefore to accelerate the training. The experimental results illustrate that the proposed method can scale up to larger batch size and further accelerate the training of ResNet, BERT and DLRM.

**Strengths:**

Strength:
1. This paper focuses on an important problem, accelerate neural network training. especially for large-batch training.
2. The proposed method is very easy to follow.
3. The authors provide some results to verify the performance of proposed method.

**Weaknesses:**

Weakness:
1. The authors can provide more visualization and analysis about the proposed method and why the proposed method can further scale the batch size. For example, the proposed method can help the model converge to s flat region?
2. The authors should provide more results about the wall time of each experiment and verify whether the proposed can save the training time.
3. I'm not sure whether you should compare your method with more baselines since LARS and LAMB is not current SOTA.

**Questions:**

N/A

---

> ### Author Rebuttal · Authors · 2023-08-08
>
> We thank the reviewer for his/her constructive suggestion. We give the response to address review's concerns step by step below.
>
> **Q1:** *The authors can provide more visualization and analysis about the proposed method and why the proposed method can further scale the batch size. For example, the proposed method can help the model converge to s flat region?*
>
> Sure, we add Fig.1 in the response PDF. Following previous discussion on generalization gap (Foret et al. 2020, Kwon et al. 2021, Zhuang et al. 2022, Zhao et al. 2022, Zhao et al. 2023, Wang et al. 2021, Simsekli et al. 2019), we add a schematic to help understanding how GSNR works to reduce generalization gap. It shows that larger GSNR helps the weights to escape from the large generalization gap area, while smaller GSNR attracts the weights to stay in small generalization gap area.
>
> We show more analysis of the reasons why our proposed method can scale the batch size:
>
> 1. **Methodological perspective.** Our proposed method provides an element-wise level of learning rate adjustment that is more accurate than existing methods and becomes more accurate when batch size gets larger. The linear scaling rule uses the same large LR for all parameters. LARS/LAMB/LANS use large LRs for the normal layers but layer-wisely or block-wisely limit LRs when $\|\theta_t\|$ is compatible with its updating quantity. VRGD that we proposed **element-wisely** limits the updating quantity for those parameters without confident gradient estimation (Fig.1b in the main context, large gradient variance or small GSNR). GSNR estimation becomes more accurate when batch size is larger. Therefore, when batch size gets extremely large, such mechanism to stabilize training may become even more accurate and helpful.
>
> 2. **Convergence rate perspective.** Applying our proposed method on basic optimizers may make the upper bound of convergence rate much tighter when increasing the batch size. For example, VR-SGD's bound depends on the lower ($r_l$) and upper bound ($r_u$) of GSNR. Larger batch size brings smaller gradient variance (eq.43 of Appendix.B) and larger GSNR (both bigger $r_l$ and $r_u$), then may result in **a tighter bound with quicker convergence** (*verified by experiments*).
>
> 3. **Generalization gap perspective.** Our proposed method can reduce more generalization gap when batch size is larger. Based on the derivations in Sec 5.2, VR-SGD has a **much smaller generalization gap** than SGD in LB training (*verified by our ImageNet experiments shown in Table.3 of the main context* ). When scaling up the batch size, such mechanism to reduce generalization becomes even more useful. Table.3 of the the main context shows that generalization gap drops 47.1% at 32k, 48.8% at 64k and 68.3% at 96k.
>
> 4. **GSNR effectiveness perspective.** The theoretical explanation of the mechanism how updating weights with smaller GSNR brings generalization gap is comprehensively discussed in previous study (Liu et al.2020). We further carried out many ablation studies in Sec.7 and found that final accuracy drops in large batch training without GSNR, which demonstrates its effectiveness in large batch scenarios.
>
> **Reference**
>
> Jinlong Liu, Guoqing Jiang, Yunzhi Bai, Ting Chen, and Huayan Wang. Understanding why neural networks generalize well through GSNR of parameters. In 8th International Conference on Learning Representations, ICLR. OpenReview.net, 2020.
>
> Foret, P., Kleiner, A., Mobahi, H., and Neyshabur, B. (2020). Sharpness-aware minimization for efficiently improving generalization. arXiv preprint arXiv:2010.01412.
>
> Kwon, J., Kim, J., Park, H., and Choi, I. K. (2021, July). Asam: Adaptive sharpness-aware minimization for scale-invariant learning of deep neural networks. In International Conference on Machine Learning (pp. 5905-5914). PMLR.
>
> Zhuang, J., Gong, B., Yuan, L., Cui, Y., Adam, H., Dvornek, N., ... and Liu, T. (2022). Surrogate gap minimization improves sharpness-aware training. arXiv preprint arXiv:2203.08065
>
> Zhao, Y., Zhang, H., and Hu, X. (2022, June). Penalizing gradient norm for efficiently improving generalization in deep learning. In International Conference on Machine Learning (pp. 26982-26992). PMLR.
>
> Ahn, K., Jadbabaie, A., and Sra, S. (2023). How to escape sharp minima. arXiv preprint arXiv:2305.15659.
>
> Wang, X., Oh, S., and Rhee, C. H. (2021). Eliminating sharp minima from SGD with truncated heavy-tailed noise. arXiv preprint arXiv:2102.04297.
>
> Simsekli, U., Sagun, L., and Gurbuzbalaban, M. (2019, May). A tail-index analysis of stochastic gradient noise in deep neural networks. In International Conference on Machine Learning (pp. 5827-5837). PMLR.
>
> **Q2:** *The authors should provide more results about the wall time of each experiment and verify whether the proposed can save the training time.*
>
> Sure, please see Table.1 and Table.2 of the response PDF. Both results show that large batch training largely reduce training time. We add them in the revision.
>
> **Q3:** *I'm not sure whether you should compare your method with more baselines since LARS and LAMB is not current SOTA.*
>
> Yes, in Table.1 of the main context, what we showed as the previous SOTA for BERT pretraining is Adasum, which performed better than LAMB. As for ImageNet, to the best of our knowledge, LARS is still the SOTA large batch optimizer. Other new method like ConAdv+AA is not included for comparison because it is not a large batch optimizer but adversarial learning instead.

---

> > ### Comment · Reviewer_xC4Q · 2023-08-20
> > **Thanks for your response!**
> >
> > Thanks for your response!
> >
> > 1. I hope to see more clear visualization about loss landscape, such as the figure in this paper [1].
> >
> > 2. Transformer model is easier to converge to sharp local minima compared with CNN. If you focus on reducing generalization gap, maybe you should provide an analysis about generalization on Transformer model, such as vision transformer or some other nlp models.
> >
> > 3. I'm happy to see the authors provide the training time results in the attached pdf. That will make the paper stronger.
> >
> > 4. I still find the performance drop of your proposed method in table 2 when scaling the batch size to 32k, for example, from 4k-77.23% to 32k-76.81%, but there is no drop from 4k to 32k for LARS. In my past experience about large batch training, 32k is not the bottleneck for imagenet training and it usually doesn't occur drop when scaling batch size to 32k. You can also find the results in original LARS paper. Could you please provide some explanation about this phenomenon. If GSNR can works well for large batch training, the performance drop should occur when the batch size is larger than 32k.
> >
> >
> >
> > [1] Visualizing the Loss Landscape of Neural Nets

---

> > > ### Author Response · Authors · 2023-08-21
> > > **We appreciate the reviewer's constructive discussion.**
> > >
> > > **Q1:** *I hope to see more clear visualization about loss landscape, such as the figure in this paper [1].*
> > >
> > > Yes, we carefully read through paper [1] and found that their Fig.2a,d showed the sharp minimum based on $L(\theta(\alpha))$, where $\theta(\alpha)=(1-\alpha)\theta+\alpha*\theta'$, $\theta$ is the optimal parameters for small batch while $\theta'$ for large batch. By taking $\alpha=[-0.5,1.5]$, they got the desired 1D loss landscape.
> > >
> > > We apologize that we didn't add such loss landscape in last rebuttal but we will try to add it in Appendix later because we can not add new PDF in the discussion period.
> > >
> > > The reason why we showed the schematic figure is that we followed many previous papers such as [2,3,4,5,6] and they just gave the schematic of generalization gap.
> > >
> > > **Reference**
> > >
> > > [1] Visualizing the Loss Landscape of Neural Nets, NIPS2018
> > >
> > > [2] Surrogate gap minimization improves sharpness-aware training. ICLR2021.
> > >
> > > [3] Penalizing gradient norm for efficiently improving generalization in deep learning. ICML 2022.
> > >
> > > [4] Efficient Sharpness-aware Minimization for Improved Training of Neural Networks, ICLR2022
> > >
> > > [5] Asam: Adaptive sharpness-aware minimization for scale-invariant learning of deep neural networks. ICML2021.
> > >
> > > [6] Towards efficient and scalable sharpness-aware minimization. CVPR2022
> > >
> > >
> > > **Q2:** *Transformer model is easier to converge to sharp local minima compared with CNN. If you focus on reducing generalization gap, maybe you should provide an analysis about generalization on Transformer model, such as vision transformer or some other nlp models.*
> > >
> > > Sure. Below shows the generalization gap reduction on BERT pretraining, which is based on transformer.  Result shows that our proposed method can also reduce the generalization gap of transformer based models by 65.7% when batch size is 64k. We will add this table into revision.
> > >
> > > |                | LAMB | VR-LAMB (ours) |
> > > |----------------|------|----------------|
> > > | Train Loss     | 1.11 | 1.31           |
> > > | Test Loss      | 1.46 | 1.43           |
> > > | Generalization Gap | 0.35 | 0.12 (-65.7%) |
> > >
> > >
> > > **Q4:** *I still find the performance drop of your proposed method in table 2 when scaling the batch size to 32k, for example, from 4k-77.23$\%$ to 32k-76.81$\%$, but there is no drop from 4k to 32k for LARS. In my past experience about large batch training, 32k is not the bottleneck for imagenet training and it usually doesn't occur drop when scaling batch size to 32k. You can also find the results in original LARS paper. Could you please provide some explanation about this phenomenon. If GSNR can works well for large batch training, the performance drop should occur when the batch size is larger than 32k.*
> > >
> > > This is because of our hyper-parameter selection strategy shown below. We didn't tune the LR until batch size reaches 64k, which means $LR=7 \cdot 2^{2}$ may not be the optimal LR and can be tuned for further improvement. [2] also used similar LR selection strategy and stated that "it is possible to achieve better results by further tuning the hyperparameters". However, the LARS results were cited from [1] and their Table.6 shows that they finetune the LR from 32k.
> > >
> > > More detailed settings were listed in Appendix.D.
> > >
> > > | Batch Size | LARS | VR-LARS (ours) |
> > > |------------|------|----------------|
> > > | 2k         | -    | $7 \times 2^{0}$          |
> > > | 4k         | -    | $7 \times 2^{0.5}$        |
> > > | 8k         | -    | $7 \times 2^{1}$         |
> > > | 16k        | -    | $7 \times 2^{1.5}$        |
> > > | 32k        | 35   | $7 \times 2^{2}$          |
> > > | 64k        | 41   | 37             |
> > > | 96k        | 43   | 38             |
> > >
> > > [1] Concurrent adversarial learning for large-batch training. ICLR 2022.
> > >
> > > [2] Large batch optimization for deep learning: Training bert in 76 minutes. ICLR 2020.

---

### Author Rebuttal · Authors · 2023-08-08

We thank the reviewers for his/her constructive suggestions. We give the response to each reviewer one by one.

We add new figures and tables in the PDF below. Please click the button below to download it.

---

### Decision · Program_Chairs · 2023-09-21

**Decision:**

Reject

**Comment:**

The paper proposes a new algorithm for large-batch training. The new algorithm offers a new variance-reduced gradient descent technique (VRGD) based on the gradient signal to noise. It applies to popular optimizers such as SGD/Adam/LARS/LAMB. The authors also provide theoretical analysis to show VRGD's superior generalization to SGD and faster convergence.  Experimental results on BERT, ResNet, and DLRM demonstrate the superiority of VRGD for large-batch training.

Since this year, many papers are around the borderline, including this one, we can only accept papers with high scores (>5.5 typically). Considering most reviewers have low intentions to accept this work, we cannot accept it. Note that the reviewer who gave a score of 7 did not provide any detailed comments, his score has a much lower weight.  We suggest the authors further improve this work according to the reviewers' comments, including more comparison with SoTA optimizers (e.g. Adan) and models (e.g. CLIP), and more intuitive explanation for the motivation and convergence and generalization superiority of the proposed method. Through reading the paper, I also found some weaknesses. So I  suggest the authors compare the first-order oracle (IFO) complexity of different optimizers in a table to show the efficiency improvement, and also use more rigorous math language to explain the generalization superiority, e.g. using a theory instead of a plain text. Fast convergence speed does not mean high efficiency, since it uses a large minibatch size for each iteration.